
# Benchmarking the Ising universality class
# in $3 \leq d < 4$ dimensions

**Claudio Bonanno**[1*]**, Andrea Cappelli**[1]**, Mikhail Kompaniets**[2,3]**,**
**Satoshi Okuda**[4] **and Kay Joerg Wiese**[5]

**1** INFN, Sezione di Firenze, Via G. Sansone 1, 50019 Sesto Fiorentino (FI), Italy
**2** Saint Petersburg State University, 7/9 Universitetskaya Embankment,
St. Petersburg, 199034, Russia
**3** Bogoliubov Laboratory of Theoretical Physics, JINR, 6 Joliot-Curie, Dubna, 141980, Russia
**4** Department of Physics, Rikkyo University Toshima, Tokyo 171-8501, Japan
**5** Laboratoire de Physique de l'École Normale Supérieure, Université PSL, CNRS,
Sorbonne Université, Université Paris-Diderot, Sorbonne Paris Cité,
24 rue Lhomond, 75005 Paris, France

⋆ claudio.bonanno@fi.infn.it

## Abstract

The Ising critical exponents $\eta$, $\nu$ and $\omega$ are determined up to one-per-thousand relative error in the whole range of dimensions $3 \leq d < 4$, using numerical conformal-bootstrap techniques. A detailed comparison is made with results by the resummed epsilon expansion in varying dimension, the analytic bootstrap, Monte Carlo and non-perturbative renormalization-group methods, finding very good overall agreement. Precise conformal field theory data of scaling dimensions and structure constants are obtained as functions of dimension, improving on earlier findings, and providing benchmarks in $3 \leq d < 4$.



## 1   Introduction

Many approaches to critical phenomena obtain results in continuous space dimension, although physically relevant dimensions are integer. Most notable is the perturbative renormalization group in $d = 4 - \epsilon$ dimensions [1–4]. This is not merely a technical issue: quantities as functions of real $d$ can clarify features that are harder to see at discrete values. E.g., one can follow the topology of the renormalization-group (RG) flow as a function of dimension and find instances where the universality class changes at non-integer values. This proved particularly useful for systems with long-range interactions [5–7] or disorder [8–13].

The recent very precise numerical conformal bootstrap [14–16] has been formulated in continuous dimension [17, 18], in particular for the Ising model in its whole range $4 > d \geq 2$ [19–21]. The interest lies in understanding how the strongly interacting Ising conformal field theory connects to a free scalar in $d = 4$ and to the integrable fully-solvable model in $d = 2$ [22, 23]. Analytic bootstrap approaches which use the dimension as a tunable parameter were also developed [24–32]. Initially, the non-unitarity of the theory in non-integer dimensions [33] was thought to hamper the numerical methods involving positive quantities. These concerns have been overcome by *de facto* never observing problems for the quantities of interest, as explained later.

In this paper, we extend the numerical approach of Ref. [20] using a single correlator, the SDPB [34] routine for determining the unitarity domain, and the Extremal Functional Method [35,36] for solving the bootstrap equations. We obtain improved results for the scaling dimensions in $4 > d \geq 3$ by a denser scanning of the unitary region near the Ising point, i.e., the kink. The latter gets parametrically sharper as $d$ approaches 4, allowing for its better identification. The conformal spectrum in dimensions $4 > d \geq 2.6$ has also been obtained in Ref. [21] via the advanced *navigator* bootstrap technique [37]. We use these very precise results in combination with ours to obtain a consistent description of the low-lying spectrum.

The achieved precision allows us to perform a detailed comparison with state-of-the-art epsilon expansion in two regimes: for $d$ close to 4, the series is directly compared to bootstrap data, using the necessary finer scale for the latter; for intermediate values between 4 and 3 (included), the divergent perturbative series is resummed using well-established methods involving the Borel transform [38–41].

The analysis is done on the dimensions of the conformal fields $\sigma, \epsilon, \epsilon'$, corresponding to spin, energy and subleading energy. They determine the critical exponents $\eta, \nu, \omega$. The precision of our bootstrap data is summarized by the (mostly) $d$-independent value of the relative error $\mathrm{Err}(\gamma)/\gamma = O(10^{-3})$ for the anomalous dimensions $\gamma$ of the conformal fields $\sigma$ and $\epsilon$. As the anomalous dimensions are very small for $d \approx 4$, the precision for the conformal dimensions $\Delta_\sigma, \Delta_\epsilon$ is even higher in this region. Regarding the subleading energy, the relative

error $\mathrm{Err}(\Delta_{\epsilon'})/\Delta_{\epsilon'}$ stays at three digits, as explained later. Some of the structure constants are determined with a higher $O(10^{-4})$ accuracy.

We compare our data with recent results of the analytic bootstrap [27–32], Monte Carlo simulations [42–44] and the non-perturbative RG [45,46]. We find that the data by all methods agree very well. This is rather rewarding given the achieved precision. Besides confirming the high quality of conformal-bootstrap results, our analysis provides a reference point for further analytic and numerical methods aiming at exploring critical phenomena in varying dimensions.

The outline of this paper is the following. In Sec. 2 we summarize our bootstrap protocol [20] and present the results for the three main conformal dimensions mentioned above, together with their polynomial fits as a function of dimension and the estimation of errors. In Sec. 3 we briefly recall the properties of the epsilon expansion and resummation techniques. We then compare its predictions with our bootstrap data and the results by other methods, and authors. A detailed analysis of all issues is presented. In Sec. 4, we report the numerical bootstrap data for scaling dimensions of structure constants and other conformal fields, and compare them to the existing epsilon expansion. In the conclusions in Sec. 5 we discuss open questions.

## 2 Conformal bootstrap in non-integer dimension

The aim of this section is to summarize our procedure for deriving conformal data of scaling dimensions and structure constants, as a function of the space-time dimension $4 > d \geq 2$. We first discuss the conformal dimensions of three main fields $\mathcal{O} = \sigma, \epsilon, \epsilon'$. Our goal is to provide a polynomial description of $\Delta_{\mathcal{O}}$ as a function of $y = 4 - d$, by performing a *best fit* of the data obtained at several values of $d$.[1] Our results are finally compared to those obtained from the resummed epsilon expansion in Section 3.

### 2.1 Summary of numerical methods

The conformal dimensions and structure constants of the critical Ising model as a function of $d$ are computed in the setup of Ref. [20], which we shortly summarize for the reader's convenience. We consider a single 4-point correlator $\langle \sigma(x_1)\sigma(x_2)\sigma(x_3)\sigma(x_4) \rangle$, where $\sigma(x)$ is the primary scalar field with lowest dimension, denoted $\Delta_{\sigma}$. We truncate the functional bootstrap equation to 190 components.[2] The unitarity condition for this equation is determined through the SDPB algorithm [34], leading to a bound in the $(\Delta_{\sigma}, \Delta_{\epsilon})$ plane; next, the Extremal Functional Method (EFM) [35, 36] is used to solve the equations on this boundary. We use the generalization of these numerical methods to non-integer dimensions developed in Ref. [20], and detailed in its Appendix A.

Our 1-correlator numerical bootstrap approach has been surpassed by more recent implementations [16, 19, 21, 47, 48], but we find it convenient for determining the low-lying spectrum with modest computing resources. The complete determination of the conformal data for one value of $d$ requires about 20 hours on 256 cores, corresponding to 5000 core hours. This simple setting allows us to evaluate the spectrum for various dimensions $d$.

The first crucial step is to locate the Ising critical point in parameter space. To this end, we adopt the twofold strategy of Ref. [20], consisting in searching the kink on the unitarity boundary in the $(\Delta_{\sigma}, \Delta_{\epsilon})$ plane and, at the same time, minimizing the central charge $c$ [15].

---

[1]Note that $\epsilon$ is the energy field, the next-to-lowest scalar primary field, not to be confused with the deviation from four dimensions denoted by $y$.

[2]This corresponds to the standard bootstrap parameter $\Lambda = 18$, which counts the number of derivatives in the approximation of the functional basis.



Figure 1: Determination of the Ising critical point for $d = 3, 3.25, 3.5, 3.75$ ($d = 3$ data from Ref. [20]). Left plots: Identification of the kink; the blue points correspond to the solutions of the bootstrap equations. Right plots: position of the $c$ minimum. The grey shaded areas represent the estimated errors on $\Delta_\sigma$, $\Delta_\epsilon$ and $c$.

This procedure allow us to determine for each value of $d$ an interval of values for $\Delta_\sigma, \Delta_\epsilon$ and $c$, that we take as the Ising conformal theory, accompanied by an estimate of the uncertainty.

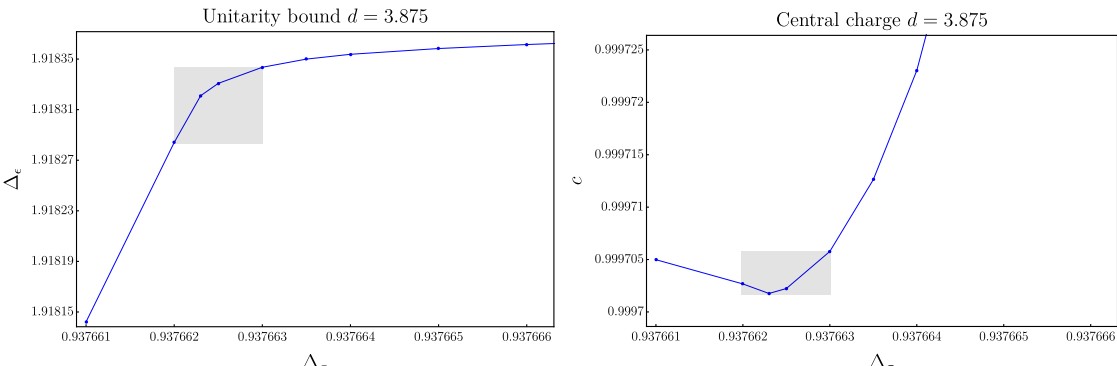

Figure 2: Determination of the Ising point for $d = 3.875$, as in Fig. 1. Note the magnified scale on both axis with respect to those of Fig. 1.

This procedure is displayed in Fig. 1, where we show the identification of the Ising point for $d = 3, 3.25, 3.5$ and $3.75$. The gray area in the plots indicates the chosen errors for $\Delta_\sigma, \Delta_\epsilon$ and $c$, which are roughly determined by the mismatch between the positions of the minimum and the kink. As a conservative choice, we consider an interval of four data points for each value of $d$.

The precision is greater than in Ref. [20], because we perform a finer scan of the $\Delta_\sigma$ values around the kink. We observe that the kink and the minimum get sharper for $d \to 4$, as shown by the four pairs of plots drawn on the same scale in Fig. 1; this is convenient in our approach, since it leads to an increased precision when anomalous dimensions are smaller. In Fig. 2, we show the point $d = 3.875$, not considered in the earlier work. It is necessary for studying the region of $d \to 4$. Here the curves are so steep that magnified scales are needed.

Once the Ising point is determined, we obtain the rest of the conformal data as follows. The solution of the bootstrap equations gives a spectrum of conformal dimensions $\Delta_\mathcal{O}$ and structure constants $f_{\sigma\sigma\mathcal{O}}$ as a function of $\Delta_\sigma$; they are divided into different sets characterized by the spin $\ell = 0, 2, 4, \ldots$ of the operator $\mathcal{O}$. The estimation of $\Delta_\mathcal{O}$ and $f_{\sigma\sigma\mathcal{O}}$ is obtained by taking the central value of such quantities for $\Delta_\sigma$ varying in the interval previously identified as the Ising point (grey areas in Figs. 1 and 2). The error is obtained from their dispersion.

It is interesting to point out that, although we largely improved the precision of our results for $4 > d > 3$ with respect to Ref. [20], we observe no signs of trouble associated to non-unitarity in our bootstrap spectrum. On general grounds, non-unitarity contributions are expected to appear for non-integer values of $d$ due to the presence of negative-norm states [33]. However, these occur at very high order in the OPE expansion of the correlator $\langle\sigma\sigma\sigma\sigma\rangle$, thus we may argue that they have numerically negligible structure constants. As a matter of fact, their presence does not seem to yield problems in solving the bootstrap equations with our method. This conclusion was also reached by recent 3-correlator bootstrap studies of the critical $O(N)$ models [18] and the Ising model [21] in non-integer space dimensions using the navigator method [37].

## 2.2 Analysis of conformal dimensions of the three leading fields for $4 > d \geq 3$

In Tab. 1 we present our results for the conformal dimensions $\Delta_\mathcal{O}$ in $4 > d > 3$ along with those of Ref. [20] for $3 \geq d > 2$, also employed in the following. Our implementation of the bootstrap determines with high precision the conformal dimensions and structure constants for the first few low-lying operators with $\ell = 0, 2$ and 4: $\mathcal{O}_{\ell=0} = \sigma, \epsilon, \epsilon'$, $\mathcal{O}_{\ell=2} = T'$ and $\mathcal{O}_{\ell=4} = C$ [20].

The goal of this section is to determine the behavior of $\Delta_\mathcal{O}$ as a function of the variable

Table 1: Conformal dimensions of the first few low-lying states for $4 > d > 2$. Exact values for $d = 2, 4$ are given in bold, results for $3 \geq d > 2$ are taken from Ref. [20].

| $d$ | $\Delta_\sigma$ | $\Delta_\epsilon$ | $\Delta_{\epsilon'}$ | $\Delta_{\epsilon''}$ | $\Delta_{T'}$ | $\Delta_C$ | $\Delta_{C'}$ |
|---|---|---|---|---|---|---|---|
| **4** | **1** | **2** | **4** | **6** | **6** | **6** | **8** |
| 3.875 | 0.9376625(5) | 1.91831(3) | 3.992(2) | 7.0(3) | 5.9307(6) | 5.8752253(9) | 7.903(3) |
| 3.75 | 0.8757175(15) | 1.83948(4) | 3.9771(12) | 6.8(2) | 5.8616(12) | 5.75111(13) | 7.81(3) |
| 3.5 | 0.753398(3) | 1.68868(5) | 3.9296(8) | 6.82(7) | 5.734(7) | 5.5053(5) | 7.55(6) |
| 3.25 | 0.633883(8) | 1.54639(9) | 3.8776(11) | 6.92(6) | 5.59(2) | 5.264(2) | 7.25(10) |
| 3 | 0.518155(15) | 1.41270(15) | 3.8305(15) | 7.01(5) | 5.505(10) | 5.026(4) | 6.7(2) |
| 2.75 | 0.40747(4) | 1.2887(2) | 3.800(2) | 7.12(8) | 5.445(15) | 4.790(5) | 6.3(2) |
| 2.5 | 0.30341(1) | 1.17625(15) | 3.7970(10) | 7.32(2) | 5.46(3) | 4.574(9) | 5.78(13) |
| 2.25 | 0.20822(3) | 1.0784(2) | 3.847(1) | 7.53(2) | 5.58(5) | 4.344(14) | 5.36(6) |
| 2.2 | 0.19053(8) | 1.0610(5) | 3.864(4) | 7.64(3) | 5.69(4) | 4.325(15) | 5.29(4) |
| 2.15 | 0.17333(8) | 1.0444(4) | 3.891(6) | 7.73(3) | 5.64(13) | 4.28(3) | 5.19(1) |
| 2.1 | 0.15663(8) | 1.0286(5) | 3.9215(5) | 7.82(3) | 5.820(10) | 4.17(4) | 5.12(4) |
| 2.05 | 0.14048(8) | 1.0134(7) | 3.9565(5) | 7.93(3) | 5.9050(10) | 4.13(6) | 5.065(15) |
| 2.01 | 0.12803(8) | 1.001(2) | 3.9900(10) | 8.035(5) | 5.9815(5) | 4.01440(10) | 5.0115(15) |
| 2.00001 | 0.125000(10) | 0.99989(14) | 4.0002(2) | 7.99(10) | 6.0006(2) | 4.000055(10) | 5.00048(8) |
| **2** | **0.125** | **1** | **4** | **8** | **6** | **4** | **5** |

$y = 4 - d$, by finding the best fitting polynomial that describes the data in Tab. 1. We use all available values, but focus on the range $4 > d \geq 3$ where results are more precise and allow for a comparison with other approaches. The points for $3 > d \geq 2$ are mainly used for stabilizing the higher powers of the fitting polynomials.[3]

We employ an improved fit method for $\Delta_\mathcal{O}(y)$ that uses orthogonal polynomials [49]: the idea is to expresses the $n^{\text{th}}$-order polynomial fit function $f_n(y)$ in terms of orthogonal polynomials $P_k(y)$ of degree $k = 0, 1, \ldots, n$, instead of a parameterization in terms of monomials, $1, y, y^2, \ldots, y^n$. To this aim we write

$$f_n(y) = \sum_{k=0}^{n} \alpha_k P_k(y), \qquad \langle P_r(y) P_s(y) \rangle \propto \sum_{i=1}^{14} P_r(y_i) P_s(y_i) \propto \delta_{rs}, \qquad (1)$$

where $y_i$ are the values in Tab. 1. This method is equivalent to the naive one, but is numerically more stable and the fit parameters $\alpha_k$ can be determined with improved precision and less statistical noise.

The optimal degree $n$ for the fitting polynomial is not known *a priori* and is determined in the following way: The fit with weights proportional to the inverse square of errors is done for several values of $n$, and the least chi-square $\chi^2_{\min}$ is found as a function of $n$. At a given order $\bar{n}$, adding a further term $\alpha_{\bar{n}+1} P_{\bar{n}+1}$ results in a negligible change of $\chi^2_{\min}$ and the best fit yields a result for $\alpha_{\bar{n}+1}$ which is compatible with zero within errors. This identifies $\bar{n}$ as the degree of the optimal polynomial. Finally, we use the results of our best fit for $\{\alpha_k\}$ to assign an error to $f_n(y)$ in the whole range of $4 > d \geq 3$. Details on the fitting procedure and the computation of errors can be found in App. A.

In this section we focus on the three leading operators $\sigma, \epsilon$ and $\epsilon'$ (corresponding to $\phi, \phi^2$ and $\phi^4$ in the $\phi^4$ field theory), which are determined with very good precision. The analysis of higher-dimensional operators is postponed to Sec. 4.2. Instead of working with conformal

---

[3]Note that the lower quality of $3 > d > 2$ data is due to the coarse scanning of $\Delta_\sigma$ values, not to an intrinsic limitation of the numerical bootstrap approach [20].

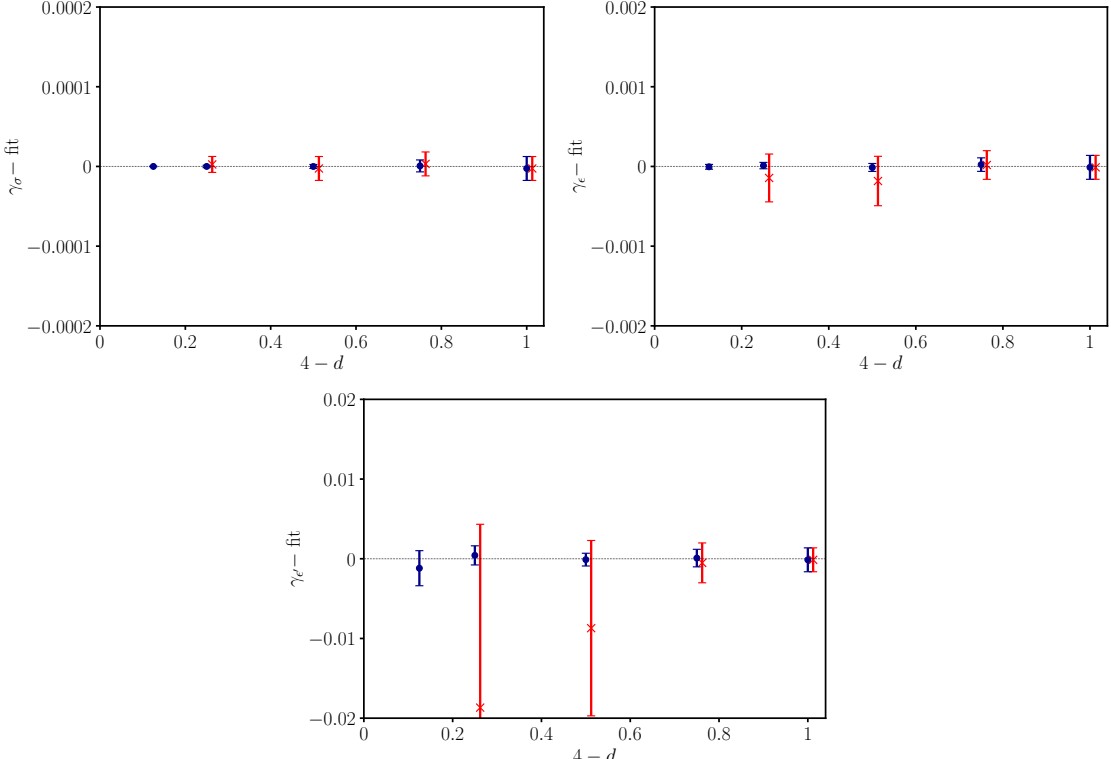

Figure 3: Old [20] (red crosses) and new (blue circles) bootstrap data for $\gamma_\sigma, \gamma_\epsilon, \gamma_{\epsilon'}$, minus the corresponding best fits. The plots use the same scales as in Ref. [20].

dimensions, we consider the anomalous dimensions

$$\gamma_\sigma = \Delta_\sigma - \frac{d-2}{2}, \qquad \gamma_\epsilon = \Delta_\epsilon - (d-2), \qquad \gamma_{\epsilon'} = \Delta_{\epsilon'} - 2(d-2). \qquad (2)$$

They are related to the Ising critical exponents $\eta$, $\nu$ and $\omega$ by

$$\eta = 2\gamma_\sigma, \qquad \frac{1}{\nu} = 2 - \gamma_\epsilon, \qquad \omega = d - 4 + \gamma_{\epsilon'}. \qquad (3)$$

The vanishing of anomalous dimensions in the free theory ($d = 4$) is assumed in the following fits.

Our analysis starts by comparing the old [20] and new data for $4 > d > 3$. In Fig. 3 the new results (blue circles) show much smaller errors than the earlier findings (red crosses), due to a more accurate localization of the Ising point, as explained above. In these and later figures we report the differences ($\gamma_{\mathcal{O}}$ − fit) between data and fitting polynomial, because simpler plots would not capture the small errors involved (note that the abscissas of the three plots differ by factors of ten). The explicit form of the best fitting polynomials are provided in Sec. 3.

Next, we compare these results with those recently obtained by solving the 3-correlator bootstrap with the navigator method [21]. In Fig. 4 our data, given in earlier figures (blue circles), are shown on a finer scale, together with the estimated error of the fit (cyan shaded area). The red triangles are the navigator values: they come with no errors and thus cannot be directly used for the fits.[4] A first observation is the fairly good agreement between the two different bootstrap approaches at our level of precision.

We propose to estimate the error of navigator data as follows. We suppose that they are roughly of the same size as those found in other 3-correlator studies at $d = 3$ (rigorous bounds)

---

[4]Earlier results of Ref. [19] are not considered here due to their large errors.

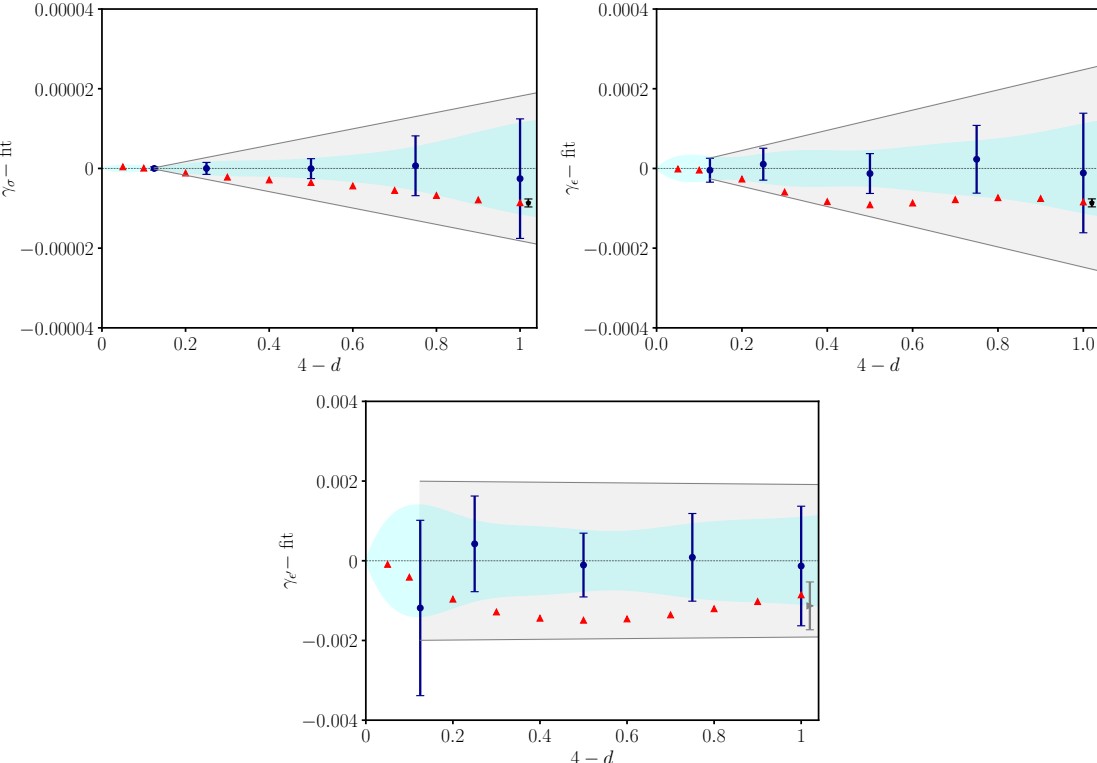

Figure 4: Plot of bootstrap data for $\gamma_\sigma, \gamma_\epsilon, \gamma_{\epsilon'}$ minus the best fit values. The shaded area represents the error obtained from the $\chi^2$ minimization of the fitting polynomial. The red triangles are results from Ref. [21] using the navigator method in a 3-correlator bootstrap setup (no error bars). Black diamonds and grey rightward triangle for $d = 3$ represent respectively results by Ref. [48] ($\gamma_\sigma$ and $\gamma_\epsilon$) and Ref. [50] ($\gamma_{\epsilon'}$); these data points are slightly displaced around $d = 3$ to improve readability. The gray shaded bands represents the error bounds reported in Eq. (4).

[48, 50], which are plotted in Fig. 4 as black diamonds ($\gamma_\sigma$ and $\gamma_\epsilon$), and a grey rightward triangle ($\gamma_{\epsilon'}$). Assuming these very small uncertainties for each value of $d$, there seems to be a negative offset with respect to our data, in particular for $\epsilon'$. This could be a systematic error due to our approximate identification of the Ising point within the unitarity region (Section 2.1), while the navigator method rigorously determines it within a unitarity island [37]. However, other explanations are possible.

In conclusion, taking into account these considerations, we enlarge the error estimate of our fits to the shaded gray bands in Figs. 4, which correspond to the following bounds:

$$\frac{\text{Err}(\gamma_\sigma)}{\gamma_\sigma} \approx \frac{\text{Err}(\gamma_\epsilon)}{\gamma_\epsilon} \lesssim 1 \times 10^{-3}, \qquad \frac{\text{Err}(\Delta_{\epsilon'})}{\Delta_{\epsilon'}} \lesssim 0.5 \times 10^{-3}, \qquad 3.875 \geq d \geq 3. \qquad (4)$$

Given the small value of anomalous dimensions for $d \to 4$, these imply extremely low absolute errors, $\text{Err}(\gamma_\sigma) = O(10^{-6})$ and $\text{Err}(\gamma_\epsilon) = O(10^{-5})$ in this range, as spelled out in the following sections. This allows us to give a precise comparison to other methods, as a benchmark for the Ising universality class in non-integer dimensions.

# 3 Comparison with the epsilon expansion in $4 > d \geq 3$

In this section, we recall some features of the epsilon expansion and the resummation methods employed for it. We compare unresummed and resummed series with the bootstrap results for $\gamma_\sigma$. Then, the analysis is extended to $\gamma_\epsilon$ and $\gamma_{\epsilon'}$.

## 3.1 Warm-up analysis of the anomalous dimensions $\gamma_\sigma$

We start with a brief summary of the properties of the perturbative expansion of the $\phi^4$ field theory in $d = 4 - y$, which describes the Ising universality class. This is a textbook subject [51] but we would like to single out a few aspects that are important in the following comparison with bootstrap results in varying dimensions.[5]

The $\beta$-function $\beta(g, y)$ and the anomalous dimensions $\gamma_{\mathcal{O}}(g)$, where $\mathcal{O} = \phi, \phi^2, \phi^4$, take the following form, in the Minimal Subtraction (MS) [51, 52] renormalization scheme,

$$\beta(g, y) = -yg + \sum_{k=2}^{n+1} \beta_k \, g^k, \qquad \gamma_{\mathcal{O}}(g) = \sum_{k=1}^{n} \gamma_{\mathcal{O},k} \, g^k. \tag{5}$$

The numerical coefficients $\beta_k, \gamma_{\mathcal{O},k}$ were computed up to order $n = 6$ in Ref. [40], and $n = 7$ in Ref. [53]. While results up to order $n = 15$ are known for a subclass of Feynman diagrams believed to give the dominant contribution, they are not used here [40, 54].

The coefficients of the $\beta$-function (5) grow exponentially with $k$, and their asymptotic behavior can be estimated from the contribution of instanton field configurations [51]

$$\beta_k \underset{k \to \infty}{\sim} C \, (-a)^k \, k^b \, k!. \tag{6}$$

Similar behaviors are found for the coefficients $\gamma_{\mathcal{O},k}$. The parameters $a, b, C$ depend on the quantity considered. One finds that the known values of the coefficients up to order $n = 7$ grow very fast with $n$ but have not yet reached their asymptotic values (6) [40, 54].

The behavior (6) can be understood as follows: The perturbative series has a vanishing radius of convergence in the complex $g$ plane, because real negative values of $g$ correspond to an upside-down potential and an action not bounded from below. This fact can be exemplified by the simple *zero-dimension path integral* (see App. B.1):

$$\mathcal{I}(g) = \int_{-\infty}^{\infty} \frac{\mathrm{d}x}{\sqrt{2\pi}} \, \mathrm{e}^{-\frac{x^2}{2} - gx^4} = \sum_{k=0}^{\infty} a_k (-g)^k, \qquad a_k = \frac{(4k)!}{2^{2k}(2k)!k!} \underset{k \to \infty}{\sim} \frac{2^{4k}}{\sqrt{2\pi k}} \times k!. \tag{7}$$

This is the generating function counting the number of vacuum Feynman diagrams. The asymptotic behavior of $a_k$ can be found by a saddle-point analysis of the integral. In field theory the corresponding saddle point is given by instantons [51].[6]

The solution of the fixed-point equation $\beta(g, y) = 0$ gives $g = g(y)$ by perturbative inversion around $g = y = 0$; this is used to rewrite the anomalous dimensions as a series in $y$,

$$\gamma_{\mathcal{O}}(y) = \sum_{k=1}^{n} \overline{\gamma}_{\mathcal{O},k} \, y^k. \tag{8}$$

This is again a divergent series of asymptotic form (6), with suitable parameters $a$, $b$ and $C$.

---

[5]An up-to-date discussion of epsilon expansion can be found in Refs. [38–41]. We refer to these works for a proof of the following statements and appropriate referencing.

[6]There is growing consensus that the large-order behavior is governed by an instanton rather than a renormalon [54]. If one could go to much higher orders in the series expansion (e.g., 20-loop order) one could apply methods of resurgence and trans-series [55].

The ratio of two consecutive terms in the series (8) can be estimated from (6) as, $\overline{\gamma}_{\mathcal{O},k}\, y/\overline{\gamma}_{\mathcal{O},k-1} \approx -aky$, which is larger than one for $y > 1/|ak|$. A simple conclusion is that the more terms are present in the perturbative series (8), the sooner it diverges as a series in $y$. We can draw two main conclusions:

i) As it stands, the perturbative series (8) is basically useless for physical dimension $y = 1$, apart from the first couple of terms, and resummation methods are necessary for extracting precise values of anomalous dimensions. The resummation is based on the Borel transform, followed by a conformal mapping, as will be explained later, and further discussed in App. B.1. This procedure gives resummed finite expressions $\widetilde{\gamma}_{\mathcal{O}}(y)$.

ii) For dimensions close to $d = 4$, i.e., $y \ll 1$, there is an optimal number of terms $n_{\mathrm{opt}}(y)$, for each $y$ value, for which the distance between the series and the resummed function $\widetilde{\gamma}_{\mathcal{O}}(y)$, $|\widetilde{\gamma}_{\mathcal{O}}(y) - \sum_{1}^{n_{\mathrm{opt}}} \overline{\gamma}_{\mathcal{O},k} y^k|$, is minimal before growing again.

The resummed anomalous dimensions $\widetilde{\gamma}_{\mathcal{O}}$ may differ from results obtained by other methods, such as the lattice formulation of the path-integral for the Ising model, or by the bootstrap. These differences are non-analytic, e.g., $\delta\gamma_{\mathcal{O}}(y) \sim \exp(-A/y)$. Within the resummation procedure, these terms may change according to how the inverse Borel transform is performed [55].

Before discussing the resummation methods in the next section, a first comparison of the perturbative expansion and the bootstrap data for $\gamma_\sigma$ clarifies the issues at stake.

The perturbative series is [40, 53]

$$
\begin{aligned}
\gamma_\sigma(y) \;=\; & 0.00925926 y^2 + 0.00934499 y^3 - 0.00416439 y^4 + 0.0128282 y^5 \\
& -0.0406363 y^6 + 0.15738 y^7, \qquad \text{(epsilon expansion)}. 
\end{aligned} \tag{9}
$$

The best polynomial fit of bootstrap data in Tab. 1 using the methods outlined in Sec. 2.2 is[7]

$$
\begin{aligned}
\gamma_\sigma(y) \;=\; & 0.009306473 y^2 + 0.008899908 y^3 - 0.001435107 y^4 + 0.001788710 y^5 \\
& -0.000533980 y^6 + 0.000128667 y^7, \qquad \text{(conformal bootstrap)}.
\end{aligned} \tag{10}
$$

The two polynomials (9) and (10) have different meanings, although their first two coefficients are close. On one hand the Feynman-diagram series is exact, but has a vanishing radius of convergence. On the other hand, the numerical bootstrap data in Tab. 1 should converge to exact non-perturbative results upon increasing the numerical precision. The collection of these values for any dimension $d = 4 - y$ gives the exact function $\gamma_\sigma^{\mathrm{ex}}(y)$, which however cannot be expressed in terms of a simple polynomial. Therefore, the fit (10) gives approximated values around $\gamma_\sigma^{\mathrm{ex}}(y)$, whose precision is *a priori* limited. Nonetheless, this description is sufficient at the present level of numerical accuracy.

In Fig. 5 we show the difference between the perturbative series (9) and the bootstrap fit (10) for $4 > d \geq 3$. Color lines correspond to the series (9) truncated at different orders $n = 2, 3, \ldots, 7$ (cf. color legend in the plot). One sees that, the higher the order $n \geq 4$, the sooner the perturbative series diverges from the bootstrap data (corresponding to the zero horizontal line in Fig. 5). The tiny errors of bootstrap points cannot be seen at this scale, thus showing that the unresummed perturbative series cannot be used for a precise determination of critical exponents in $d = 3$, as stated in point *ii)* above. Yet, the lower terms $n = 2, 3$ may provide crude estimates. Fig. 6 shows the other regime, close to four dimensions. Only the

---

[7]Note that the best-fit polynomial (10) starts with an $O(y^2)$ term, because the linear term vanishes within errors. If a linear term is included in the fit procedure, it leads to a coefficient three orders of magnitude smaller than the quadratic term. Therefore, the conformal bootstrap implies $\gamma_\sigma(y) = O(y^2)$ close to $d = 4$, in agreement with perturbation theory.

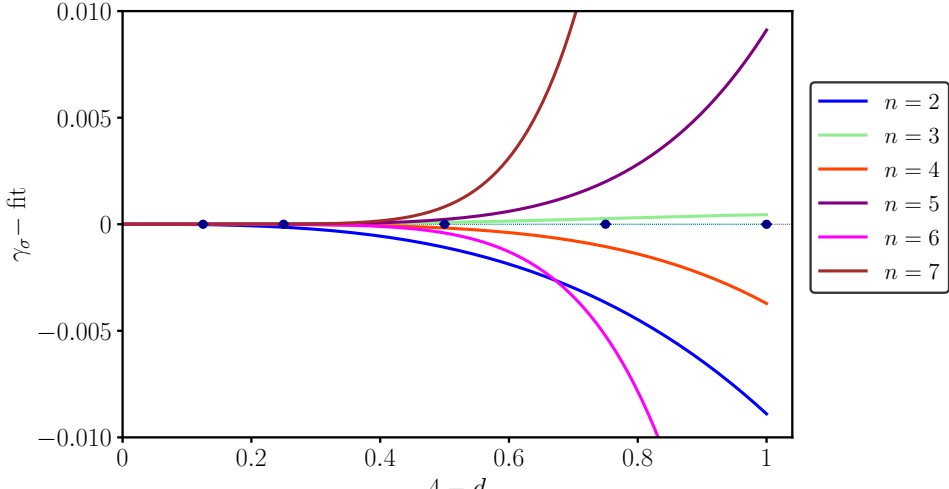

Figure 5: Comparison of $\gamma_\sigma$ bootstrap data with unresummed epsilon expansion (9) in the region $4 > d > 3$ for truncations of the series to order $n = 2, \ldots, 7$ (see color legend). All quantities have been subtracted by the best fit values (see (10)).

bootstrap point for $d = 3.875$ is present in this range, but we also show results of Ref. [21] for $d \geq 3.8$, which match very well while lacking error bars, as discussed earlier.[8] In contrast to the $d \approx 3$ region, we observe that the truncated perturbative series shows a different behavior. At any given $y$ value, upon increasing the perturbative order up to an optimal value $n_{\mathrm{opt}} \sim 1/y$, the perturbative series approaches the zero horizontal line (with a cyan error band), before starting to diverge. Namely, it matches the exact bootstrap value $\gamma_\sigma^{\mathrm{ex}}(y)$, within numerical errors.

Therefore, the comparison between non-perturbative bootstrap results and unresummed epsilon expansion for $\gamma_\sigma(y)$ is extremely good in the region $4 > d > 3.8$, with precision $\mathrm{Err}(\gamma_\sigma) \approx 1 \times 10^{-6}$, i.e., $\mathrm{Err}(\gamma_\sigma)/\gamma_\sigma < 1 \times 10^{-3}$. According to the previous discussion, we conclude that we do not see any non-perturbative difference for $d \to 4$.

We remark that the epsilon expansion can also be obtained by analytic solution of the bootstrap equations around $d = 4$, assuming a perturbative expansion near the free theory [24, 25, 27, 28, 30–32]. Thus, is our comparison in Fig. 6 tautological? It is not, because the bootstrap identity is a set of consistency conditions that depends on the kind of quantities they act on. Our numerical solution does not assume any perturbative expansion, i.e., it is an independent solution of the bootstrap constraints. That without any perturbative input, our conformal bootstrap results accurately reproduce perturbative predictions close to $d = 4$ is non-trivial.

A natural question is how our numerical bootstrap approach can reproduce the perturbative series, i.e., in which regime the two polynomials (9) and (10) may agree beyond the $O(y^3)$ term. As said earlier, the bootstrap polynomial (10) is approximated, it can at most describe a band of values around $\gamma_\sigma^{\mathrm{ex}}(y)$. While the size $\mathrm{Err}(\gamma_\sigma)$ of this band stays finite in the whole range $0 < y < 1$ (see plots), that of the epsilon expansion is expanding in $y$ and can be finite only for $y < y_{\mathrm{max}} \sim O(1/n)$, $n$ being the perturbative order. We expect that, upon running the bootstrap for several points $y_i$, with $0 < y_i < y_{\mathrm{max}} \ll 1$, and by performing best fits with polynomials limited to such a small interval, one may find that the two expressions (9) and (10) match order by order, i.e., the epsilon expansion is fully recovered.

---

[8]Note that the red triangles are not used in our fit of bootstrap data.

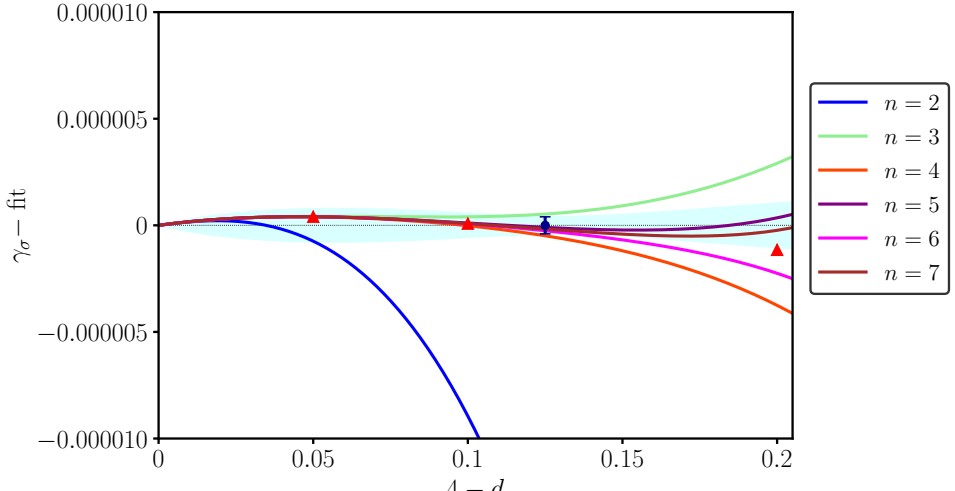

Figure 6: Comparison of $\gamma_\sigma$ data minus best fit in the region $4 > d > 3.8$, between bootstrap (blue circle) and unresummed epsilon expansion (9) with different truncations of the perturbative series (cf. Fig. 5). The red triangles are the results of the bootstrap navigator method [21]. The cyan shaded area is the fit error.

## 3.2 Bootstrap data versus resummed perturbative results

Precise estimates of the critical exponents have been obtained over the years by refining the resummation techniques applied to the epsilon expansion series [2–4, 40, 41, 51, 56, 57]. In this work, we use the methods of Refs. [40, 41] extended to dimension $4 > d \geq 3$. Let us briefly recall the main steps involved [51]. The Borel transform $\mathcal{B}_{\gamma_\mathcal{O}}(t)$ of the perturbative expansion for the anomalous dimension $\gamma_\mathcal{O}$ (8) is defined by removing the factorial growth from the series,

$$\mathcal{B}_{\gamma_\mathcal{O}}(t) = \sum_{k=1}^{n} \frac{\overline{\gamma}_{\mathcal{O},k}}{k!} t^k. \tag{11}$$

One infers from the asymptotic behavior (6) that this function has a singularity $\mathcal{B}_{\gamma_\mathcal{O}}(t) \sim (1 + ta)^{-b-1}$ and a corresponding finite radius of convergence.

The resummed quantity is defined by the inverse Borel transform,

$$\widetilde{\gamma}_\mathcal{O}(y) = \int_0^\infty \mathrm{d}t \, e^{-t} \, \mathcal{B}_{\gamma_\mathcal{O}}(yt). \tag{12}$$

By definition $\gamma_\mathcal{O}(y)$ in (8) and $\widetilde{\gamma}_\mathcal{O}(y)$ in (12) have the same perturbative expansion; however, the latter should be better behaved if $\mathcal{B}_{\gamma_\mathcal{O}}(t)$ is suitably continued analytically outside the original disc $|t| < 1/|a|$ to a region including the real positive axis.[9] Such analytic continuation in principle requires the knowledge of all singularities of $\mathcal{B}_{\gamma_\mathcal{O}}(t)$ in the complex $t$-plane. At this point, one can only make educated guesses on these singularities, that translate into (physical) ansatzes for $\widetilde{\gamma}_\mathcal{O}(y)$.

In practice, one assumes that the only singularity of $\mathcal{B}_{\gamma_\mathcal{O}}(t)$ lies at $t = -1/a$ real and negative, and that it is a branch cut extending to $t = -\infty$. Using a conformal mapping $t(z)$, this branch cut is mapped onto the unit circle, with the start of the branch cut mapped onto $z = -1$, and $t = -\infty$ to $z = 1$, preserving the origin $z = t = 0$. As long as there are no other singularities, $\mathcal{B}(t(z))$ has a radius of convergence one in $z$. As $t = \infty$ corresponds to $z = 1$,

---

[9]In particular, a real negative value of the parameter $a$ in (6), i.e., a perturbative series (8) of definite sign, is problematic.

Table 2: Conformal dimensions of $\sigma, \epsilon$ and $\epsilon'$ field from resummed perturbative expansion, obtained according to the methods of [40].

| $d$ | $\Delta_\sigma$ | $\Delta_\epsilon$ | $\Delta'_\epsilon$ |
|---|---|---|---|
| 3.875 | 0.937662197(7) | 1.91831086(14) | 3.9924550(11) |
| 3.75 | 0.8757158(3) | 1.839419(4) | 3.97529(3) |
| 3.5 | 0.753393(10) | 1.68854(7) | 3.9276(5) |
| 3.25 | 0.63386(8) | 1.5458(4) | 3.873(2) |
| 3 | 0.5181(3) | 1.4108(12) | 3.820(7) |

this allows one to perform the inverse Borel transform (12). Details on this procedure can be found in App. B.1.

This general idea can be improved in several ways, allowing one to introduce a set of free parameters. The latter are determined such that the final result is the least sensible to their variation. Apart from providing a robust resummation scheme, the parameter uncertainty implies an estimate of the resummation error. These methods have been improved over the years by taking into account the phenomenology of critical phenomena [51]. In our work, the resummed data are obtained by extending the setup of Refs. [38, 40, 41] from $d = 3$ to non-integer dimensions. A complete account of these methods is too long to be presented here; nonetheless, we provide some introductory material that will allow the reader to assess the original works. In App. B.1, the resummation is worked out in a toy model, where one can compare it with the exact result. In App. B.2, instead, a "reader's guide" to Ref. [40] is presented, together with the values of the resummation parameters used here.

Let us also mention that another option for the analytic continuation is to use Hypergeometric functions, for which the inverse Borel transform can be written as a Meijer-G function [56]. One drawback of this approach is the possibility for spurious poles on the integration contour. As here we could not give justice to their influence, we exclude this resummation method.

Figure 7 shows the fitted bootstrap data (blue points) of $\gamma_\sigma(y)$ already reported in Fig. 4, now compared to the resummed epsilon-expansion values of Tab. 2 (green squares).[10] The agreement between these two results is very good, especially for $d \geq 3.5$, where the unresummed series (magenta line) is already diverging, and greatly improves on earlier studies [2,3] analyzed in [20]. Let us remark that resummed $\widetilde{\gamma}_\sigma(y)$ values have been obtained for non-integer dimensions down to $d = 2$, still finding agreement with bootstrap data, although with larger uncertainties. Finally, Fig. 7 shows the latest Monte Carlo results in $d = 3$ (yellow rhombus), that match extremely well the bootstrap points. Further $d = 3$ results by these and other methods are summarized in a later figure. Finally, Fig. 7 and later plots for the dimensions $\gamma_\varepsilon$ and $\gamma_{\varepsilon'}$ also report a solid red curve linearly interpolating the navigator points of Ref. [21] obtained for $4 > d \geq 3$. This allows one to assess the negligible difference between the two sets of bootstrap data in the comparison to the epsilon-expansion.

We now extend the previous analysis to the energy field $\epsilon$. The best fit of the conformal bootstrap data is

$$
\begin{aligned}
\gamma_\epsilon(y) = \ & 0.333441601y + 0.114095325y^2 - 0.083458310y^3 \\
& + 0.081381007y^4 - 0.045296977y^5 + 0.014290102y^6 \\
& - 0.001741325y^7 , \qquad \text{(conformal bootstrap)} .
\end{aligned}
\tag{13}
$$

---

[10]Resummations in this section use the 6-loop results, that were verified in several independent works [40,41, 53]. We do not use the 7-loop results of Ref. [53], since they were not yet checked independently. Past experience, e.g., with the 5-loop results, teaches us that involved perturbative calculations require confirmation.

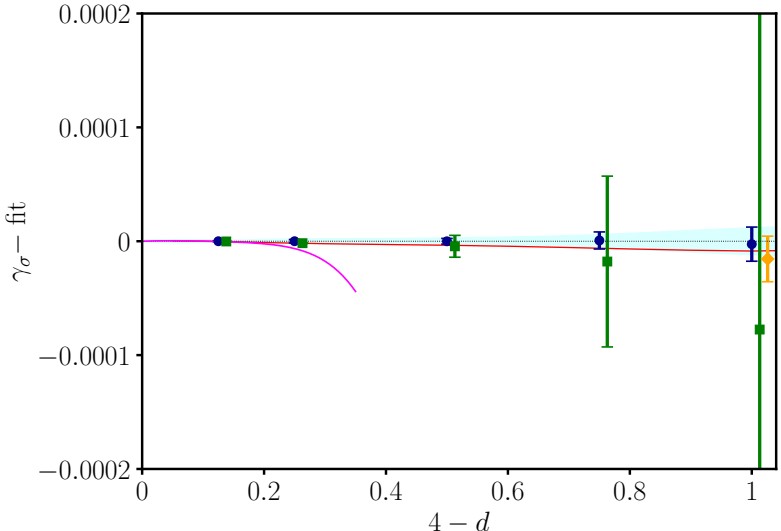

Figure 7: Comparison of $\gamma_\sigma$ data minus best-fit values: bootstrap (blue circles), Borel-resummed epsilon expansion [40] (green squares), unresummed high-order epsilon expansion (magenta solid curve), $d = 3$ Monte Carlo [44] (yellow rhombus). We also plot a solid red line linearly interpolating results of Ref. [21] for $4 > d \geq 3$. Note that data points are slightly displaced around the same $d$ values ($d = 3.875$, $d = 3.75$, $d = 3.5$, $d = 3.25$ and $d = 3$) to improve readability. Results from earlier work [3] have been omitted due to their large error bars.

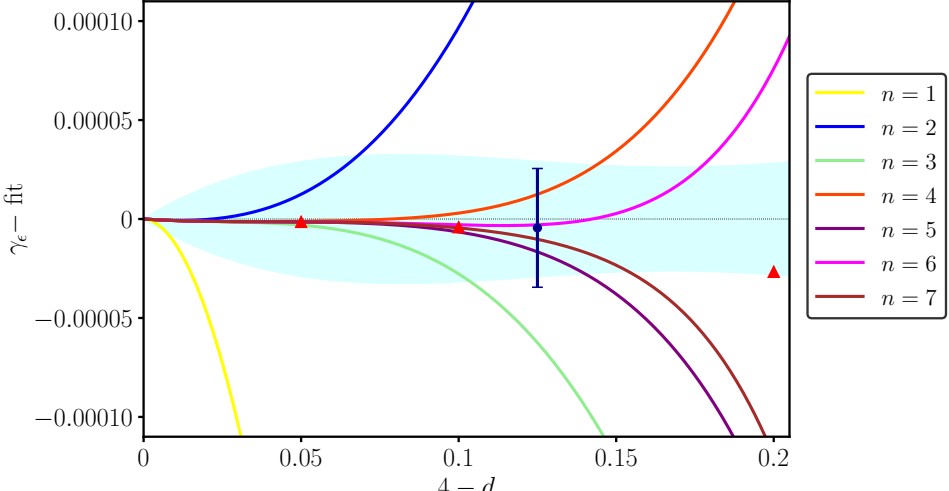

Figure 8: Comparison of the $\gamma_\epsilon$ data minus the best fit in the region $4 > d > 3.8$. Our bootstrap point is the blue circle with error bar; the triangles are obtained by the navigator method [21]; the different truncations of the perturbative series are as in Fig. 5. The cyan shaded area is the fit error.

The epsilon-expansion series reads [40, 53]

$$
\begin{aligned}
\gamma_\epsilon(y) =\ & 0.333333y + 0.117284y^2 - 0.124527y^3 + 0.30685y^4 - 0.95124y^5 \\
& + 3.57258y^6 - 15.2869y^7, \qquad \text{(epsilon expansion)}.
\end{aligned} \tag{14}
$$

One remarks the agreement, within errors, of the first two coefficients of this series; this corrects less precise results of [20] (cf. Fig. 6b there).

Table 3: Conformal dimension of $\epsilon$ field from resummed perturbative expansion, obtained according to the methods of [41].

| $d$ | $\Delta_\epsilon$ |
|---|---|
| 3.9 | 1.93440534057(12) |
| 3.8 | 1.8706742(6) |
| 3.7 | 1.808546(5) |
| 3.6 | 1.747876(2) |
| 3.5 | 1.68858(6) |
| 3.4 | 1.63062(15) |
| 3.3 | 1.5740(3) |
| 3.2 | 1.5187(5) |
| 3.1 | 1.4647(9) |
| 3 | 1.4122(15) |

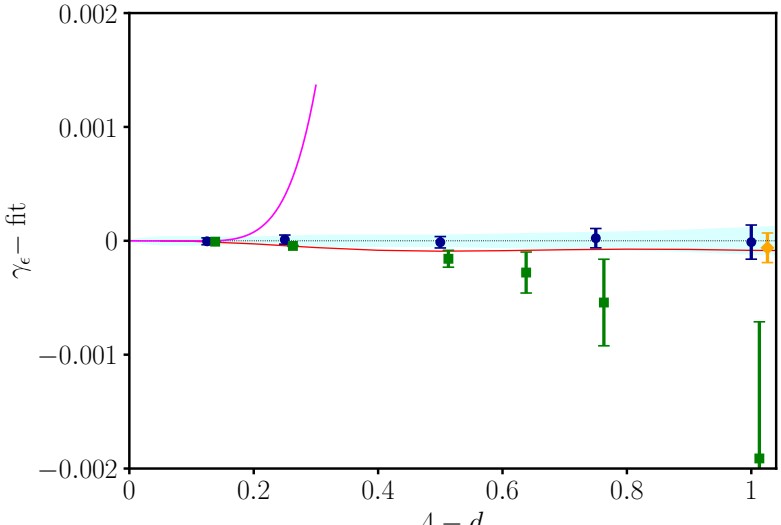

Figure 9: Comparison of $\gamma_\epsilon$ data minus best fit: bootstrap (blue circles), Borel-resummed epsilon expansion [40] (green squares), unresummed epsilon expansion (magenta solid curve), $d = 3$ Monte Carlo [44] (yellow rhombus). We also plot a solid red line linearly interpolating results of Ref. [21] for $4 > d \geq 3$. The cyan shaded area is the fit error as in earlier plots.

The comparison for $d \rightarrow 4$ before resummation is shown in Fig. 8. As for Fig. 7, the truncated perturbative series for $\gamma_\epsilon$ are plotted. Their curves approach the bootstrap fit (horizontal zero axis with cyan error band) with better and better precision. Note the remarkable quality of the navigator method (red triangles) [21]. Altogether, the agreement for $d \rightarrow 4$ is found with high precision, $\text{Err}(\gamma_\epsilon) = 3 \times 10^{-5}$ and $\text{Err}(\gamma_\epsilon)/\gamma_\epsilon = 1 \times 10^{-3}$.

Figure 9 presents a comparison with the resummed perturbative series (Tab. 2): the agreement is again very good for $4 > d \geq 3.5$; there is a small $O(10^{-3})$ deviation from the bootstrap and Monte Carlo results [44] (yellow rhombus) in $d = 3$. Probably there is a slight underestimation of the error. Let us remark that this resummation procedure is *honest*, as it does not use the exact $d = 2$ conformal dimension as an input, with which it could be improved. The

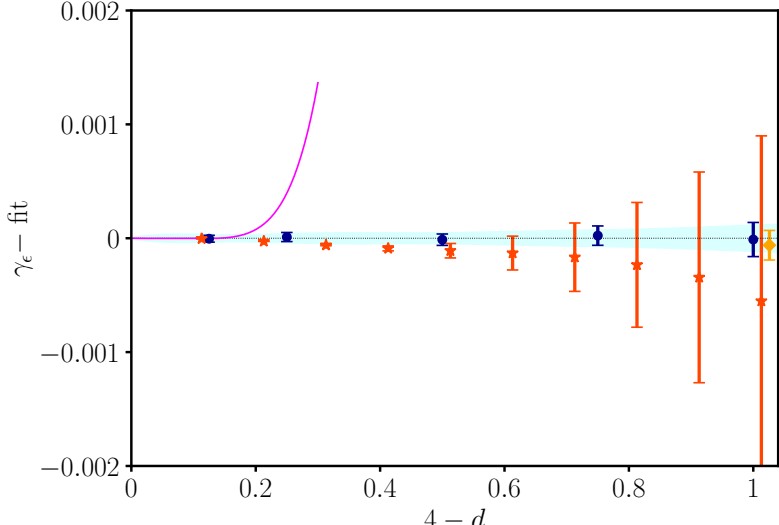

Figure 10: Comparison of $\gamma_\epsilon$ minus best fit: bootstrap (blue circles), Self-Consistent resummed epsilon expansion [41] (red stars), unresummed epsilon expansion (magenta solid curve), $d = 3$ Monte Carlo [44] (yellow rhombus).

comparison with another method, called Self-Consistent (SC) resummation[11] is presented in Fig. 10, where we plot data of Tab. 3. In this case, the Borel transform is done on the perturbative series of $1/v^3$, instead of $1/v = 2 - \gamma_e$: this choice is motivated by a match with the $d = 2$ conformal field theory, that is achieved through comparing the $n$ dependence of the $O(n)$-symmetric $\phi^4$ theory [41]. We conclude that adding information of the exact results in $d = 2$ improves the resummation of the perturbative series (for this particular critical exponent). A similar constraint does not seem to be possible for the other critical exponents, as discussed in Ref. [41].

Summarizing, the bootstrap and epsilon-expansion results agree very well: for $d \to 4$ the unresummed series fits perfectly, for $4 > d \geq 3$ there is remarkable agreement, keeping in mind that the resummation error is roughly one order of magnitude larger than that of bootstrap and Monte Carlo results.

A comparison of all $d = 3$ results available in the literature for $\gamma_\sigma$ and $\gamma_\epsilon$ is given in Figs. 11 and 12. The corresponding numerical values are in Tab. 4. Besides data already discussed (drawn in earlier colors), we report recent results of the non-perturbative renormalization group [45] (brown downward triangle). The central value is given by our fit of the bootstrap data with error given by the cyan band, not by the mean value of all results. The Figs. 11 and 12 respect our convention of plotting the two anomalous dimensions on scales differing by one order of magnitude, roughly equal to the ratio of their actual value. Finally, Tab. 4 and Figs. 11, 12 report also the results of other 3-correlator bootstrap approaches, using EFM [48] and the navigator method [50], and paying particular attention to error estimates (cf. rigorous bounds). We also remark that the results obtained by perturbative expansions directly in $d = 3$ [3,4] are consistent with bootstrap results too, but have one order of magnitude larger errors and are therefore not plotted in Figs. 11 and 12.

We now analyze the subleading $\mathbb{Z}_2$-even scalar field $\epsilon'$, which is related to the critical

---

[11]See Ref. [41] for a detailed discussion of this approach.

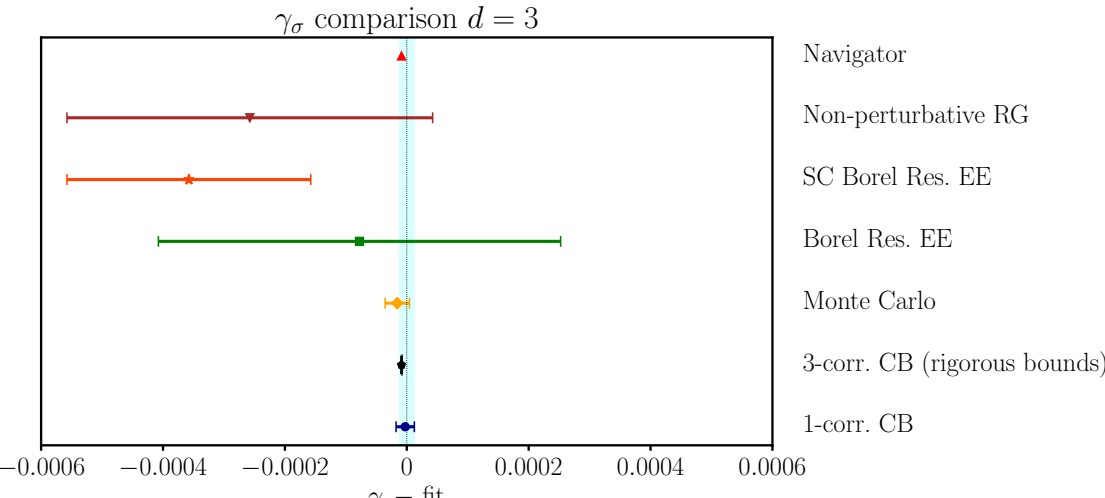

Figure 11: Summary of up-to-date predictions for $\gamma_\sigma$ at $d = 3$ (minus best fit): 1-correlator bootstrap [20] (blue circle), 3-correlator bootstrap with rigorous bounds [48] (black pentagon), Monte Carlo [44] (yellow rhombus), Borel-resummed epsilon expansion [40] (green square), Self-Consistent resummed epsilon expansion [41] (red star), non-perturbative renormalization group [45] (brown downward triangle), bootstrap navigator method [21] (red upward triangle).

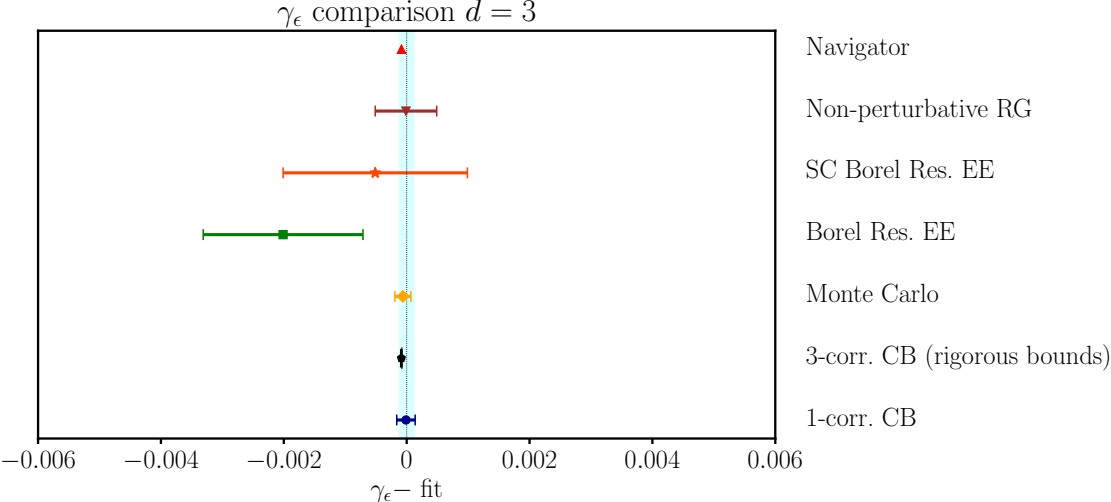

Figure 12: Summary of up-to-date predictions for $\gamma_\epsilon$ in $d = 3$ (minus best fit): 1-correlator bootstrap [20] (blue circle), 3-correlator bootstrap with rigorous bounds [48] (black pentagon), Monte Carlo [44] (yellow rhombus), Borel-resummed epsilon expansion [40] (green square), Self-Consistent resummed epsilon expansion [41] (red star), non-perturbative renormalization group [45] (brown downward triangle), bootstrap navigator method [21] (red upward triangle).

exponent $\omega = \Delta_{\epsilon'} - d = d - 4 + \gamma_{\epsilon'}$. The best fit of our data gives:[12]

$$
\begin{aligned}
\gamma_{\epsilon'}(y) = \ & 2.000178549y - 0.518006835y^2 + 0.721996645y^3 \\
& -0.684437170y^4 + 0.447648598y^5 - 0.162903635y^6 \\
& +0.026155257y^7, \qquad \text{(conformal bootstrap)}.
\end{aligned}
\tag{15}
$$

---

[12]The fit again assumes $\gamma_{\epsilon'} = 0$ for $d = 4$.

Table 4: Comparison of $d = 3$ results for the conformal dimensions of low-lying fields: 1-correlator bootstrap [20], 3-correlator bootstrap [48] (errors on $\Delta_\sigma$ and $\Delta_\epsilon$ are rigorous bounds), Borel-resummed epsilon expansion [40], Self-Consistent (SC) Borel-resummed epsilon expansion [41], Monte Carlo [42,44], non-perturbative renormalization group [45,46] and bootstrap navigator method with rigorous bounds [50].

| $d = 3$ Ising critical indices | $\Delta_\sigma$ | $\Delta_\epsilon$ | $\Delta_{\epsilon'}$ |
|---|---|---|---|
| Bootstrap (1 correlator) | 0.518155(15) | 1.41270(15) | 3.8305(15) |
| Bootstrap (3 correlators) | 0.5181489(10) | 1.412625(10) | 3.8297(2) |
| Borel resummed epsilon expansion | 0.5181(3) | 1.4107(13) | 3.820(7) |
| SC Borel resummed epsilon expansion | 0.5178(2) | 1.4122(15) | 3.827(13) |
| Monte Carlo | 0.51814(2) | 1.41265(13) | 3.832(6) |
| Non-perturbative RG | 0.5179(3) | 1.41270(50) | 3.832(14) |
| Navigator (rigorous bounds) | 0.518157(35) | 1.41265(36) | 3.8295(6) |

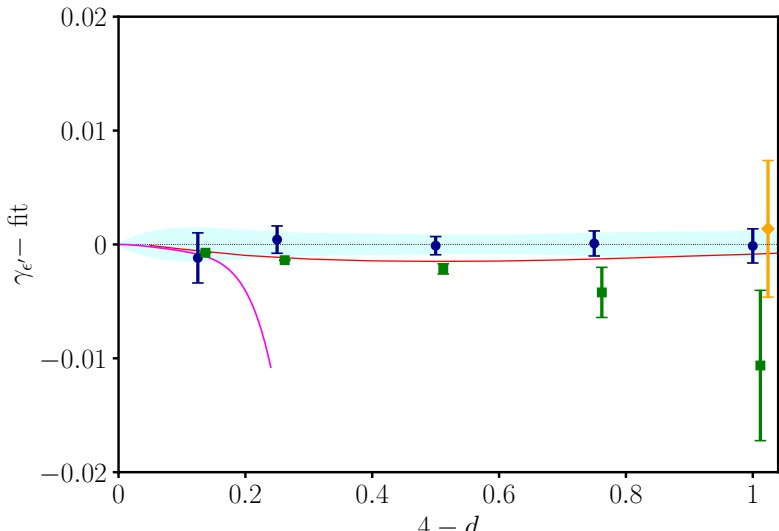

Figure 13: Comparison of $\gamma_{\epsilon'}$ data minus best fit: bootstrap (blue circles), Borel-resummed epsilon expansion [40] (green squares), unresummed epsilon expansion (magenta solid curve), $d = 3$ Monte Carlo [42] (yellow rhombus). We also plot a solid red line linearly interpolating results of Ref. [21] for $4 > d \geq 3$.

The large errors of the earlier analysis [20] have been reduced, as explained earlier (see Fig. 3). The epsilon-expansion series is [40,53],

$$
\begin{aligned}
\gamma_{\epsilon'}(y) &= 2y - 0.62963y^2 + 1.61822y^3 - 5.23514y^4 + 20.7498y^5 \\
&\quad - 93.1113y^6 + 458.7424y^7, \qquad \text{(epsilon expansion)}.
\end{aligned} \tag{16}
$$

In Fig. 13 we show the difference between the data and the bootstrap best fit (15). The overall error of the fit for $\gamma_{\epsilon'}$ is estimated to be less than $2.0 \times 10^{-3}$ in the whole range. The relative error is $\text{Err}(\gamma_{\epsilon'})/\gamma_{\epsilon'} = 1 \times 10^{-3}$ for $d = 3$ but goes up to[13] $1 \times 10^{-2}$ for $d = 3.875$. The comparison with Monte Carlo [42,44] in $d = 3$, and the resummed epsilon-expansion series are also shown, finding again good agreement at the coarser scale (note a factor of 10

---

[13]The growth of the error when passing from $d = 3.75$ to $d = 3.875$ is due to the instability of the higher part of the spectrum when approaching $d = 4$. This issue is further discussed in Sec. 4.3.

**Sci**Post

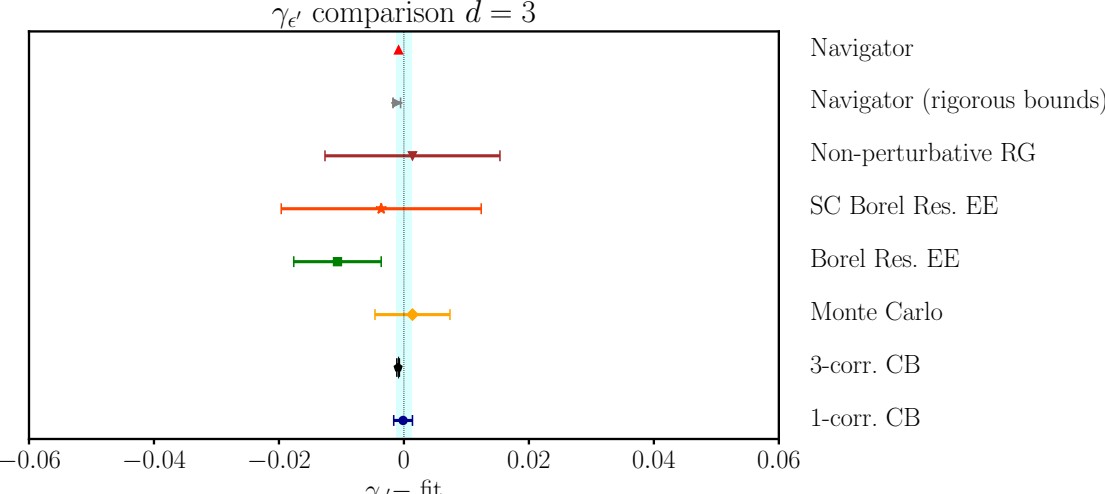

Figure 14: Summary of up-to-date predictions for $\gamma_{\epsilon'}$ in $d = 3$ (minus our best fit, from bottom to top): 1-correlator bootstrap [20] (blue circle), 3-correlator bootstrap [48] (black pentagon), Monte Carlo [44] (yellow rhombus), Borel-resummed epsilon expansion [40] (green square), Self-Consistent resummed epsilon expansion [41] (red star), non-perturbative renormalization group [45] (brown downward triangle), bootstrap navigator method with rigorous bounds [50] (grey rightward triangle), bootstrap navigator method [21] (red upward triangles).

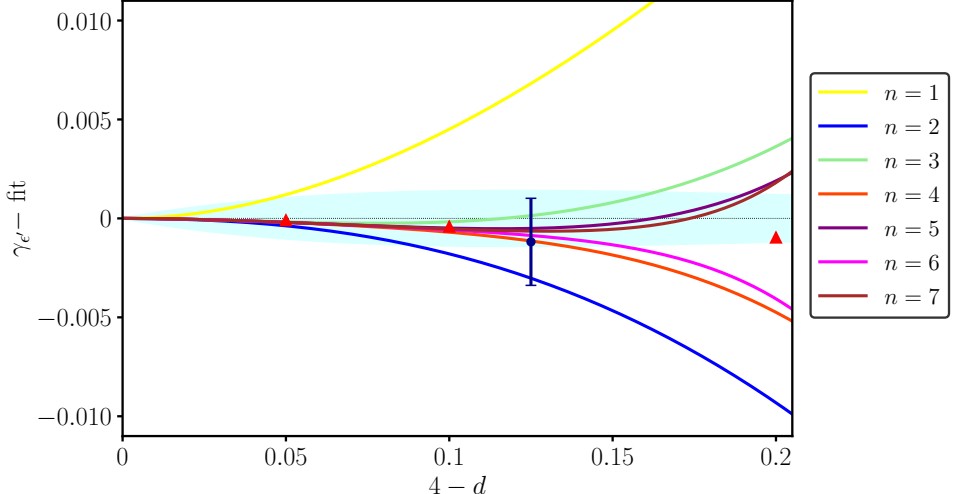

Figure 15: Comparison of the $\gamma_{\epsilon'}$ data minus the best fit in the region $4 > d > 3.8$. Our bootstrap point is the blue circle with error bar; the triangles are obtained by the navigator method [21]; the different truncations of the perturbative series are as in Fig. 5. The cyan shaded area is the fit error.

w.r.t. Fig. 9). A systematic difference between bootstrap and epsilon-expansion points is seen for $d \to 3$, similar to what was found for $\gamma_\epsilon$ in Fig. 9. Such a drift is smaller for the navigator results [21] (red line) than for our data, for $4 > d \geq 3.5$. Further values of $\Delta_{\epsilon'}$ in $d = 3$ found in the literature are reported in Tab. 4 and plotted in Fig. 14. A zoom over the region close to $d = 4$ is drawn in Fig. 15, showing the same features as in Figs. 6 and 8.

We conclude this section by stressing the very good overall agreement of bootstrap and

Table 5: Structure constants of the first few low-lying states for $4 > d > 2$. The exact values for $d = 2, 4$ are given in bold, results for $3 \geq d > 2$ are taken from [20].

| $d$ | $c$ | $f_{\sigma\sigma\epsilon}$ | $f_{\sigma\sigma\epsilon'}$ | $f_{\sigma\sigma\epsilon''} \times 10^4$ | $f_{\sigma\sigma T'}$ | $f_{\sigma\sigma C}$ | $f_{\sigma\sigma C'}$ |
|---|---|---|---|---|---|---|---|
| **4** | **1** | **1.4142136** | **0** | **0** | **0** | **0.169031** | **0** |
| 3.875 | 0.99970(2) | 1.38228(2) | 0.015298(14) | 0.33(10) | 0.003070(2) | 0.1540603(3) | 0.000772(2) |
| 3.75 | 0.998594(3) | 1.34586(3) | 0.027517(15) | 1.4(3) | 0.005641(5) | 0.133(8) | 0.00134(10) |
| 3.5 | 0.9922615(15) | 1.26132(3) | 0.04426(3) | 4.0(2) | 0.00911(10) | 0.105(5) | 0.0021(3) |
| 3.25 | 0.976864(6) | 1.16282(4) | 0.05225(3) | 6.0(3) | 0.0106(2) | 0.084(6) | 0.0019(9) |
| 3 | 0.946535(15) | 1.05184(4) | 0.05300(5) | 7.1(4) | 0.010575(15) | 0.065(5) | 0.0020(5) |
| 2.75 | 0.893275(15) | 0.92939(4) | 0.04794(8) | 7.0(4) | 0.00901(6) | 0.048(4) | 0.00235(15) |
| 2.5 | 0.807110(10) | 0.796303(5) | 0.03885(2) | 5.90(9) | 0.00668(3) | 0.033(3) | 0.0029(3) |
| 2.25 | 0.677724(2) | 0.65311(2) | 0.02738(4) | 4.27(5) | 0.00394(14) | 0.0195(15) | 0.0035(2) |
| 2.2 | 0.64609(7) | 0.62333(6) | 0.0245(5) | 3.76(9) | 0.00352(7) | 0.019(4) | 0.0038(3) |
| 2.15 | 0.61243(8) | 0.59313(8) | 0.0225(5) | 3.36(2) | 0.0025(5) | 0.017(3) | 0.00385(15) |
| 2.1 | 0.57680(10) | 0.56249(7) | 0.02018(8) | 2.98(7) | 0.00265(5) | 0.016(3) | 0.00395(15) |
| 2.05 | 0.53935(15) | 0.53143(8) | 0.01785(5) | 2.58(4) | 0.00230(10) | 0.0135(25) | 0.00390(10) |
| 2.01 | 0.5082(3) | 0.5058(6) | 0.01605(5) | 2.246(9) | 0.00193(3) | 0.01550(10) | 0.003920(10) |
| 2.00001 | 0.500015(15) | 0.499998(5) | 0.015623(4) | 2.0(2) | 0.0018520(5) | 0.0148235(15) | 0.0039040(10) |
| **2** | **0.5** | **0.5** | **0.0156250** | **2.1972656** | **0.00185290** | **0.0148232** | **0.003906** |

resummed epsilon expansion. The study in varying dimensions clarifies the different behavior of quantities in the perturbative and non-perturbative regimes.

# 4 Structure constants and scaling dimensions of higher fields

In this section we analyze further bootstrap data. The structure constants (OPE coefficients) of low-lying fields $\sigma, \epsilon, \epsilon', T$ are very precise, the error being on the fifth decimal, thus better than those of the corresponding conformal dimensions presented earlier. Next we discuss subleading and spinful fields, $\epsilon'', T', C, C'$, presenting results for both dimensions and structure constants. Some of them are good, others are not completely correct, showing the limits of our numerical bootstrap approach.

## 4.1 Structure constants in $4 > d \geq 3$

Tab. 5 reports all data for structure constants: those for $4 > d > 3$ are new results, the ones for $3 \geq d > 2$ are taken from [20]. The central charge $c$ is obtained from the structure constant $f_{\sigma\sigma T}$ of the energy-momentum tensor $T$ by

$$f_{\sigma\sigma T}^2 = \frac{d}{4(d-1)} \frac{\Delta_\sigma^2}{c}. \tag{17}$$

For $f_{\sigma\sigma\mathcal{O}}$, we adopt the by-now standard normalization of [21, 48]. The relation with the earlier normalization $\widetilde{f}_{\sigma\sigma\mathcal{O}}$ of Ref. [15] is

$$f_{\sigma\sigma\mathcal{O}}^2 = \frac{\left(\frac{d-2}{2}\right)_\ell}{(d-2)_\ell} \widetilde{f}_{\sigma\sigma\mathcal{O}}^2, \tag{18}$$

where $(x)_\ell \equiv \Gamma(x+\ell)/\Gamma(x)$ is the Pochhammer symbol.

The central charge $c$ and the structure constants $f_{\sigma\sigma\epsilon}$ and $f_{\sigma\sigma\epsilon'}$ are determined with very high accuracy: their dependence on $y = 4 - d$ is obtained with the fit method of Sec. 3.1, assuming the exact $d = 4$ value. The resulting polynomials are reported together with the

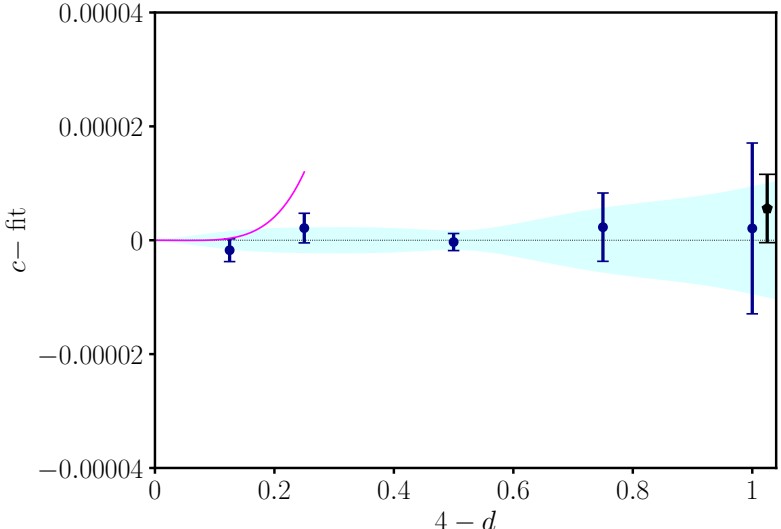

Figure 16: Comparison of $c$ data minus best fit: bootstrap (blue circles), unresummed epsilon expansion [58, 59] (magenta solid curve), 3-correlator bootstrap at $d = 3$ [48] (black pentagon).

available epsilon-expansion series [30, 31, 58, 59]:

$$
\begin{aligned}
c(y) \;=\;& 1 - 0.015415049 y^2 - 0.026663929 y^3 - 0.004992140 y^4 - 0.010357094 y^5 \\
& + 0.007424814 y^6 - 0.004670278 y^7 + 0.001206599 y^8 , \\
& \hspace{5cm} \text{(conformal bootstrap)} , \hspace{1cm} (19)
\end{aligned}
$$

$$
\begin{aligned}
c(y) \;=\;& 1 - 0.0154321 y^2 - 0.0266347 y^3 \\
& - 0.0039608 y^4 , \hspace{3cm} \text{(epsilon expansion)} , \hspace{1cm} (20)
\end{aligned}
$$

$$
\begin{aligned}
f_{\sigma\sigma\epsilon}(y) \;=\;& \sqrt{2} - 0.235465537 y - 0.170275458 y^2 + 0.096635030 y^3 - 0.113371408 y^4 \\
& + 0.100586943 y^5 - 0.054667196 y^6 + 0.016161292 y^7 - 0.001992399 y^8 , \\
& \hspace{5cm} \text{(conformal bootstrap)} , \hspace{1cm} (21)
\end{aligned}
$$

$$
\begin{aligned}
f_{\sigma\sigma\epsilon}(y) \;=\;& \sqrt{2} - 0.235702 y - 0.168047 y^2 + 0.103680 y^3 - 0.224776 y^4 , \\
& \hspace{4cm} \text{(epsilon expansion)} , \hspace{1cm} (22)
\end{aligned}
$$

$$
\begin{aligned}
f_{\sigma\sigma\epsilon'}(y) \;=\;& 0.136221303 y - 0.118250195 y^2 + 0.067116467 y^3 - 0.058700794 y^4 \\
& + 0.037159615 y^5 - 0.012211017 y^6 + 0.001647332 y^7 , \\
& \hspace{5cm} \text{(conformal bootstrap)}, \hspace{1cm} (23)
\end{aligned}
$$

$$
f_{\sigma\sigma\epsilon'}(y) \;=\; 0.1360828 y + 0.11844240525 y^2 , \hspace{1cm} \text{(epsilon expansion)} . \hspace{1cm} (24)
$$

We remark: *i*) the excellent agreement between the first few terms of the conformal bootstrap and epsilon-expansion series, and *ii*) the need of a high-order $O(y^7, y^8)$ polynomial for precise fits. The corresponding curves are shown in Figs. 16 and 17. Note that $c$, $f_{\sigma\sigma\epsilon}$ and $f_{\sigma\sigma\epsilon'}$ were determined with strikingly small (relative) errors, respectively $O(10^{-5})$, $O(10^{-4})$ and $O(10^{-4})$ over the entire $d$ range.

The comparison with other conformal bootstrap results is as follows: The best 3-correlator determination in $d = 3$ [48] is shown as a black pentagon in the figures. Data from the navigator method are unfortunately only available for $f_{\sigma\sigma\epsilon}$ [21]. The agreement among different

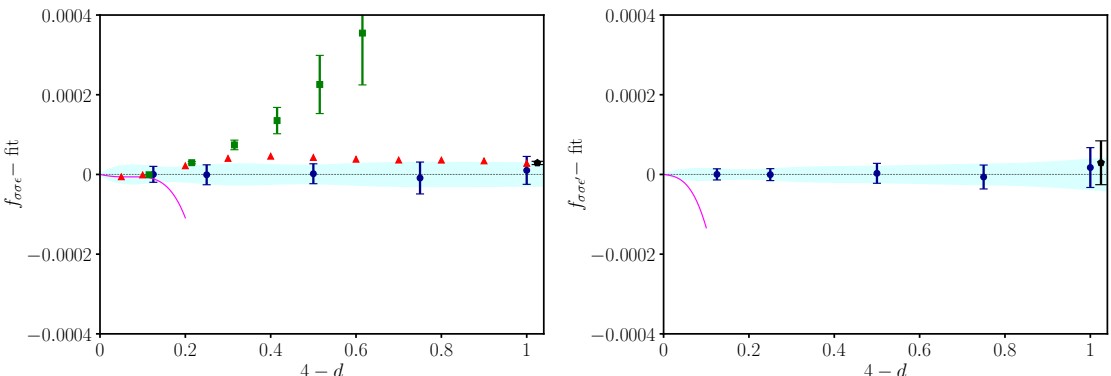

Figure 17: Comparison of $f_{\sigma\sigma\epsilon}$ and $f_{\sigma\sigma\epsilon'}$ minus best fit: bootstrap (blue circles), unresummed epsilon expansion [30, 31, 58, 59] (magenta solid curve), 3-correlator bootstrap at $d = 3$ [48] (black pentagon). For $f_{\sigma\sigma\epsilon}$ we also report the resummed epsilon expansion (green squares) and bootstrap navigator results [21] (red triangles).

Table 6: Structure constant $f_{\sigma\sigma\epsilon}$ from resummed perturbative expansion, obtained according to the methods of [41].

| $d$ | $f_{\sigma\sigma\epsilon}$ |
|---|---|
| 3.9 | 1.3890497(2) |
| 3.8 | 1.360960(3) |
| 3.7 | 1.330222(12) |
| 3.6 | 1.29703(3) |
| 3.5 | 1.26154(7) |
| 3.4 | 1.22386(13) |
| 3.3 | 1.1841(2) |
| 3.2 | 1.1423(3) |
| 3.1 | 1.0986(5) |
| 3 | 1.0531(7) |

numerical setups is extremely good. Moreover, as already observed for scaling dimensions, the unresummed epsilon expansion captures the $d \to 4$ behavior, and it does it very well, since the lower-order terms of the respective polynomials (19)–(24) are equal within errors. For $f_{\sigma\sigma\epsilon}$, the results of the resummed epsilon expansion, reported in Tab. 6, are also shown, determined by earlier methods: the 4$^{\text{th}}$-order series (22) only allows for a precise agreement down to $d \approx 3.6$, given the fine scale of Fig. 17. For the remaining quantities, the epsilon expansion is either too short for a resummation, or not alternating.

## 4.2 Higher fields $T'$ and $C$

The analysis of the fields $T'$ ($\ell = 2$) and $C$ ($\ell = 4$) is done along the same lines. The fit polynomials for $\Delta_{T'}$ and $\Delta_C$, obtained as before, are

$$
\begin{aligned}
\Delta_{T'}(y) = {} & 6 - 0.567900778y + 0.1779633663y^2 - 0.806164966y^3 \\
& + 1.749534636y^4 - 1.684842086y^5 + 0.765011179y^6 \\
& - 0.126284231y^7, \qquad\qquad \text{(conformal bootstrap)}, \qquad (25) \\
\Delta_C(y) = {} & 6 - 1.001598184y + 0.030791232y^2 \\
& - 0.033868719y^3 + 0.041665026y^4 - 0.002907562y^5 \\
& - 0.006602770y^6, \qquad\qquad \text{(conformal bootstrap)}. \qquad (26)
\end{aligned}
$$



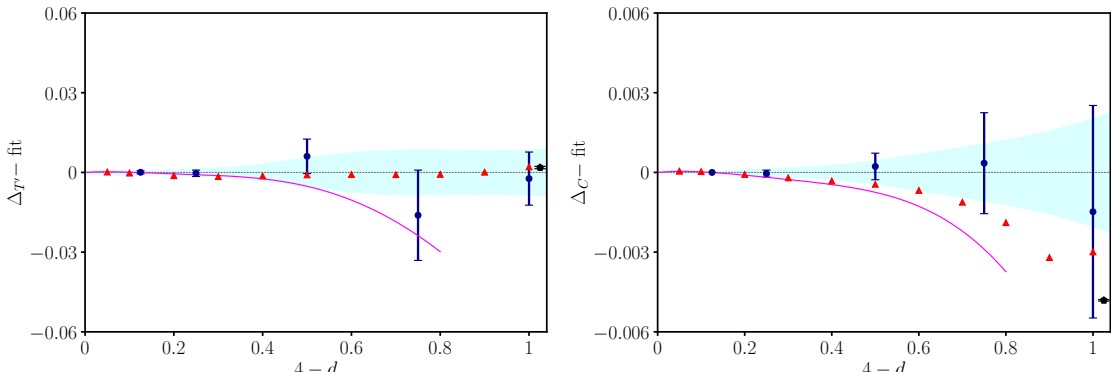

Figure 18: Comparison of scaling dimensions minus best fit for $T', C$ fields: bootstrap (blue round points), navigator method [21] (triangle red points), 3-correlator bootstrap at $d = 3$ [48] (black pentagon) and unresummed epsilon expansion [27, 32, 58, 59] (magenta solid line).

They are shown in Fig. 18, along with the bootstrap results of [21] (red triangles) and the available epsilon-expansion series (magenta solid lines) [27, 32, 58, 59]:

$$\Delta_{T'}(y) = 6 - 0.5555556y, \qquad \text{(epsilon expansion)}, \qquad (27)$$
$$\Delta_C(y) = 6 - y + 0.01296296y^2 + 0.01198731y^3$$
$$- 0.006591585y^4, \qquad \text{(epsilon expansion)}. \qquad (28)$$

As shown by the cyan band, representing our fitting error, the scaling dimensions of these fields are determined with an accuracy comparable to that achieved for the low-lying $\ell = 0$ states: $\text{Err}(\Delta_{T'}) \approx 10^{-2}$ and $\text{Err}(\Delta_C) \approx 3 \times 10^{-3}$, meaning that $\text{Err}(\Delta_{T'})/\Delta_{T'} \approx 10^{-3}$ and $\text{Err}(\Delta_C)/\Delta_C \approx 5 \times 10^{-4}$. Within our precision, we observe very good agreement with the results of [21] (especially for $T'$). Furthermore, the unresummed epsilon expansion is again in agreement with the bootstrap results for $d \to 4$. Overall, the picture is consistent with the $\ell = 0$ case discussed earlier.[14]

The corresponding structure constants are given by the polynomial fits

$$f_{\sigma\sigma T'}(y) = 0.026278214y - 0.012019512y^2 - 0.016779681y^3$$
$$+ 0.025762223y^4 - 0.018571573y^5 + 0.006902659y^6$$
$$- 0.001000504y^7, \qquad \text{(conformal bootstrap)}, \qquad (29)$$
$$f_{\sigma\sigma C}(y) = 0.16903085 - 0.122480930y + 0.077087613y^2 - 0.591032947y^3$$
$$+ 1.331591787y^4 - 1.231373513y^5 + 0.512308476y^6$$
$$- 0.079520247y^7, \qquad \text{(conformal bootstrap)}. \qquad (30)$$

They can be compared to the available epsilon expansions [27, 32, 58–60]:

$$f_{\sigma\sigma T'}(y) = 0.02635231y - 0.013176155y^2, \qquad \text{(epsilon expansion)}, \qquad (31)$$
$$f_{\sigma\sigma C}(y) = 0.16903085 - 0.12244675y + 0.02131741y^2$$
$$+ 0.002168567y^3 - 0.0019760553y^4, \qquad \text{(epsilon expansion)}. \qquad (32)$$

The comparison is shown in Fig. 19. Also in this case we observe good agreement between the conformal bootstrap polynomials and the epsilon expansion series up to $O(y^3)$ terms.

---

[14]The good behavior of the perturbative expansion for larger values of $y \approx 0.8$ is not stressed, since it may be an artifact of the low order of the series.

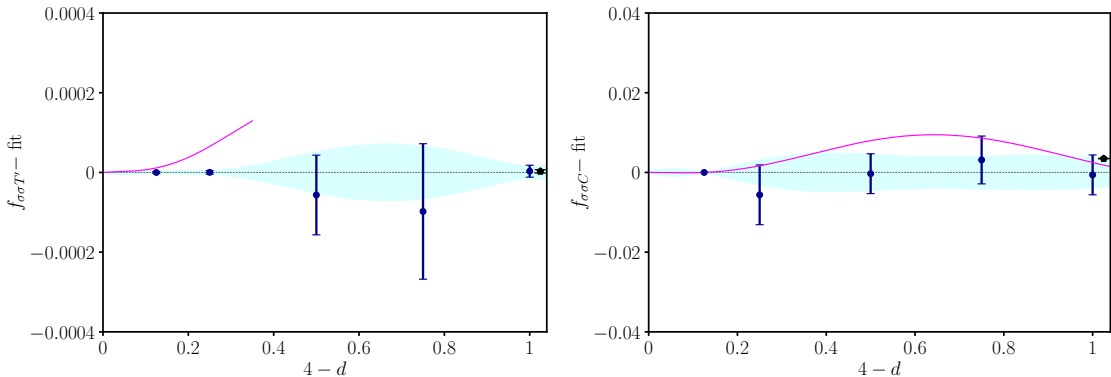

Figure 19: Behavior of structure constants $f_{\sigma\sigma T'}$ and $f_{\sigma\sigma C}$ (round blue points) compared with 3-correlator bootstrap at $d = 3$ [48] (black pentagon) and epsilon expansion (magenta solid line) [58, 59].

## 4.3 Subleading fields $\epsilon''$ and $C'$

The numerical 1-correlator bootstrap approach used in this paper is known to have a limited precision for states higher up in the conformal spectrum, in particular for our approximation to 190 components of the truncated bootstrap equations. In this section, we show that our identification of $\epsilon''$ ($\ell = 0$) and $C'$ ($\ell = 4$) has some problems, especially for $d \to 4$. We explain these difficulties by using the epsilon expansion for conformal dimensions and structure constants, as well as the 3-correlator bootstrap data [21] in varying dimensions, which are definitely more accurate for the higher spectrum than our results. We think that these aspects are worth discussing, especially because the $y = 4 - d$ dependence plays a crucial role.

We start our analysis from the subleading twist $\ell = 4$ operator $C'$, for which we find the following best fit polynomial:

$$\Delta_{C'}(y) = 8 - 0.827053961y - 0.055211344y^2 + 0.053430207y^3$$
$$+ 0.010354264y^4 - 0.003205703y^5, \qquad \text{(conformal bootstrap)}. \qquad (33)$$

These data are shown in Fig. 20 (left part). It turns out that $C'$ is degenerate at $d = 4$ with another field with same dimension and spin, called $C'_2$. Their dimensions are known to leading order in the epsilon expansion,

$$\Delta_{C'}(y) = 8 - 1.555556y, \qquad (34)$$
$$\Delta_{C'_2}(y) = 8 - 0.833333y, \qquad \text{(epsilon expansion)}, \qquad (35)$$

and are plotted in Fig. 20 with magenta dashed and solid lines, respectively. Near these lines, the navigator bootstrap results [21] are plotted with gold and red triangles.

One sees that our results start at $d \to 4$ very close to $C'_2$ (see first coefficient in polynomials (33) and (34)) and end up near $C'$ at $d = 3$. Therefore, the state we found is a mixture of $C'$ and $C'_2$: better numerical precision would be needed for disentangling the two states near $d \to 4$, obtained, e.g., by increasing the number of components approximating the bootstrap equations.

The fit of the structure constant is given by

$$f_{\sigma\sigma C'}(y) = 0.006871047y - 0.005215834y^2 - 0.003223129y^3$$
$$+ 0.005087571y^4 - 0.001393464y^5, \qquad \text{(conformal bootstrap)}, \qquad (36)$$

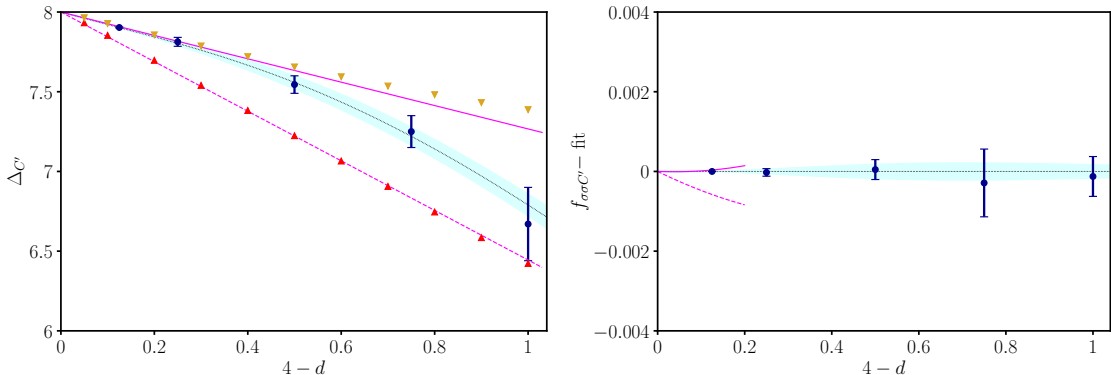

Figure 20: Scaling dimension and structure constant of would-be $C'$ operator in our bootstrap spectrum (blue circles). Upward red and downward gold triangles represent navigator results for $C'$ and $C'_2$ [21]. The dashed and solid magenta lines are the corresponding leading-order epsilon expansion.

and plotted in the right part of Fig. 20. The epsilon-expansion results for $C'$ and $C'_2$ read,

$$f_{\sigma\sigma C'}(y) = 0.001543806y\,, \tag{37}$$
$$f_{\sigma\sigma C'_2}(y) = 0.006458202y\,, \qquad \text{(epsilon expansion)}\,, \tag{38}$$

and are shown as magenta dashed and solid lines on the right of Fig. 20.

These perturbative data show a remarkable fact: for $d < 4$ the state of higher dimension $C'_2$ has a larger structure constant, contrary to the standard behavior of $f_{\sigma\sigma\mathcal{O}}$ decreasing fast with $\Delta_{\mathcal{O}}$. It is thus clear that, close to $d = 4$, $C'_2$ gives the dominant contribution to a putative mixed $C'$-$C'_2$ state. This suggests the reason why our results with limited precision start close to $C'_2$. The analysis is confirmed by the bootstrap result for the structure constant in (36): for $d \to 4$ it fits the perturbative behavior of $f_{\sigma\sigma C'_2}$, as seen in the right plot of Fig. 20. In conclusion, our subleading $\ell = 4$ state is identified as $C'_2$ for $d \to 4$, but gradually approaches $C'$ in $d = 3$.

Another problematic identification concerns the $\epsilon''$ field (corresponding to $\phi^6$ in the $\phi^4$ theory). The best fit of bootstrap data gives

$$\Delta_{\epsilon''}(y) = 2.313321845y - 1.678645012y^2 + 0.336440006y^3$$
$$+ 0.090959178y^4\,, \qquad \text{(conformal bootstrap)}\,, \tag{39}$$

while the leading epsilon-expansion result reads [24, 25, 60]:

$$\Delta_{\epsilon''}(y) = 2y - 4.759259y^2\,, \qquad \text{(epsilon expansion)}\,. \tag{40}$$

For the structure constant we find

$$f_{\sigma\sigma\epsilon''}(y) = 0.002851280y^2 - 0.003188068y^3 + 0.001218496y^4$$
$$- 0.000161879y^5\,, \qquad \text{(conformal bootstrap)}\,, \tag{41}$$
$$f_{\sigma\sigma\epsilon''}(y) = 0.006901444y^2\,, \qquad \text{(epsilon expansion)}\,. \tag{42}$$

It is apparent that our bootstrap results do not match the leading perturbative expansion for $d \to 4$. The corresponding plots are shown in Fig. 21, where the disagreement with bootstrap results from Ref. [21] (red triangles) is also seen.

Let us investigate the possibility of another mixing of states near $d \to 4$. In this case there is no degenerate field with $\epsilon''$ at $d = 4$. However, the next subleading one $\epsilon''' \sim \Box^2 \phi^4$ in

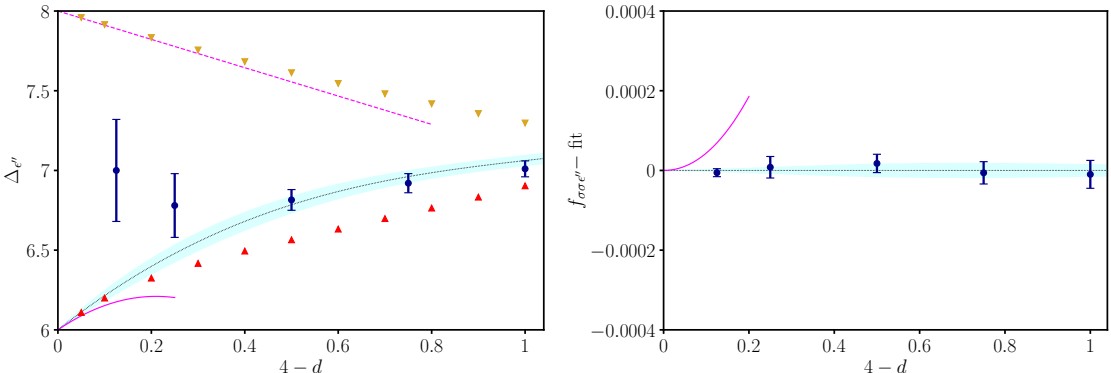

Figure 21: Scaling dimension and structure constant of the would-be $\epsilon''$ operator in our bootstrap spectrum (blue circles). Upward red and downward gold triangles represent navigator results for $\epsilon''$ and $\epsilon'''$ [21]. The solid and dashed magenta lines are the corresponding leading-order epsilon expansion, which agree with the navigator results, but not ours.

the $\phi^4$ theory is present at higher dimension $\Delta_{\epsilon'''} \leq 8$. The epsilon expansion and navigator results for this field are also shown in Fig. 21 (left part, gold downward triangles). We remark that a mixing of $\epsilon''$ and $\epsilon'''$ was shown to take place at $d = 2.8$, i.e., rather far from $d = 4$ [21].

We suppose that the limited resolution of our data finds a state which is a mixture of $\epsilon''$ and $\epsilon'''$ also for $d \to 4$, but we cannot be certain of this. As for $C'$ and $C_2'$, support for this argument could come from a comparison of the corresponding structure constants $f_{\sigma\sigma\epsilon''}$ and $f_{\sigma\sigma\epsilon'''}$. Unfortunately, the epsilon expansion of the latter is not available, so we cannot get a definite explanation of our $\Delta_{\epsilon''}$ data.

## 5  Conclusions

In this paper we obtained the conformal dimensions and structure constants of the critical Ising CFT as a function of varying dimension $4 > d \geq 3$ by using the numerical conformal bootstrap approach.

Our main result is the precise determination of the anomalous dimensions of the $\sigma, \epsilon, \epsilon'$ fields, which are related to the Ising critical exponents $\eta$, $\nu$, $\omega$. Our relatively simple 1-correlator bootstrap setup is able to compute the $d$-dependence of these quantities with up to one-per-thousand relative accuracy; therefore, our findings can be used as a benchmark for future studies in non-integer space dimension. For these low-lying states of the conformal spectrum, our results are in very good agreement with those of more advanced 3-correlator bootstrap techniques [21,37,47,48], with a small offset included in the error estimate.

We presented a detailed comparison of available predictions from different methods. For $d \to 4$, our results agree with those from unresummed perturbation theory. This shows two things: that non-perturbative differences, which might effect the bootstrap program or the resummed series, are negligible for $d \to 4$. The other non-trivial result is that both approaches agree on the same analytic continuation in dimension. A possible explanation of this correspondence is provided by the analytical bootstrap, which on one hand reproduces the epsilon expansion, and on the other hand uses the same ingredients as the numerical bootstrap.

For $3 \leq d < 4$, but away from $d = 4$, the bootstrap data agree very well with other results, obtained by resummation techniques of the perturbative series, Monte Carlo simulations, and other bootstrap approaches. In the whole $4 > d \geq 3$ range we find overall consistency among

the different approaches; improvements are needed by adding further terms to the perturbative series in $d = 3$, as the current state of the art still shows a $O(10^{-3})$, $O(10^{-2})$ discrepancy, respectively for $\nu$ and $\omega$, and in general much larger error bars than bootstrap and Monte Carlo results.

We were able to compute bootstrap data for the conformal dimensions of higher-order fields in $4 > d \geq 3$, including the lowest-lying spinful fields $T'$ ($\ell = 2$) and $C$ ($\ell = 4$), with a precision comparable to that of spinless operators. The central charge and OPE coefficients of low-lying fields were obtained with even higher precision than that of the corresponding anomalous dimensions. The structure constants agree well with those of the 3-correlator boot-strap, where available (mostly in $d = 3$), and with perturbation theory for $d \to 4$.

A possible future development is to improve current bootstrap results in the region $3 > d \geq 2$, in order to better understand how the $d = 3$ theory approaches the $d = 2$ Virasoro minimal model. To this aim, it is important to go beyond the lowest-lying states and precisely probe higher-dimensional and higher-spin fields. Improved 3-correlator bootstrap protocols, such as the recently proposed navigator method, may be well suited here.

## Acknowledgements

We are grateful to C. Bonati, R. Guida, J. Henriksson, S. Kousvos, L. Maffi, R. Pisarski, M. Reehorst, S. Rychkov, M. Serone and B. Sirois for useful discussions. We thank Insitut Pascal for organizing the workshop "Bootstat", where this work was initiated. CB acknowledges the support of the Italian Ministry of Education, University and Research under the project PRIN 2017E44HRF, "Low dimensional quantum systems: theory, experiments and simulations". Numerical computations have been performed on the `Zefiro` cluster of the Scientific Computing Center at INFN Pisa.

## A  Orthogonal polynomial regression

Standard polynomial regression of the data set $S \equiv \{x_i, y_i, \Delta y_i\}_{i=1}^N$ is achieved by minimizing

$$\chi^2 = \sum_{k=1}^{N} \left( \frac{y_k - f(x_k)}{\Delta y_k} \right)^2, \tag{A.1}$$

with respect to the parameters $\{c_i\}_{i=0}^d$ of the fit function,

$$f_n(x) = \sum_{r=0}^{n} c_r x^r. \tag{A.2}$$

The degree $n$ of the polynomial is not known a priori.

A smarter fit is obtained by changing the basis in which the polynomial is expressed:

$$\mathcal{B}_{\text{naive}} = \left\{ 1, x, x^2, \ldots, x^d \right\} \to \mathcal{B}_{\text{ortho}} = \left\{ P_0(x), P_1(x), P_2(x), \ldots, P_d(x) \right\}, \tag{A.3}$$

where the polynomials $P_k(x)$ (of degree $k$) are chosen to be *orthogonal* on the independent variables of the dataset $S$, i.e.:

$$\langle P_r(x) P_s(x) \rangle_S = \frac{1}{N} \sum_{k=1}^{N} P_r(x_k) P_s(x_k) = k_r^2 \delta_{rs}, \tag{A.4}$$

where $k_r$ are constants. With this choice, the fit function becomes

$$f_n(x) = \sum_{r=0}^{n} \alpha_r P_r(x). \tag{A.5}$$

The best fit is obtained by minimizing $\chi^2$ in Eq. (A.1 ). The advantage of the orthogonal polynomial regression is that the coefficients $\alpha_r$ do not depend on the $\alpha_s$ with $s > r$, i.e., adding higher-degree polynomials $r > n$ to $f_n(x)$ does not change the value of $\alpha_r$ with $r \leq n$ within the statistical errors [49]. Thus, this procedure is better suited to assess the optimal degree of the polynomial.

The expression of the polynomials $P_r(x)$ is known in the literature. In this work, we follow the conventions of Ref. [49]. We start by fixing the $r = 0$ and $r = 1$ polynomials as

$$P_0(x) = 1, \qquad P_1(x) = 2(x - a_1), \qquad a_1 = \frac{1}{N} \sum_{k=1}^{N} x_k \equiv \overline{x}. \tag{A.6}$$

Higher-order polynomials with $r \geq 2$ are obtained through the recursive relation [49],

$$P_{r+1}(x) = 2(x - a_{r+1})P_r(x) - b_r P_{r-1}(x), \tag{A.7}$$

where the coefficients $a_{r+1}$ and $b_r$ are given by

$$a_{r+1} = \frac{\sum_{k=1}^{N} x_k P_r^2(x_k)}{\sum_{k=1}^{N} P_r^2(x_k)}, \qquad b_r = \frac{\sum_{k=1}^{N} P_r^2(x_k)}{\sum_{k=1}^{N} P_{r-1}^2(x_k)}. \tag{A.8}$$

In this work, we find the best fitting polynomial for $\gamma_{\mathcal{O}}$ and $f_{\sigma\sigma\mathcal{O}}$ as a function of $y = 4 - d$. We always assume their known analytic value for $d = 4$, for example $\gamma_{\mathcal{O}}(d = 4) = 0$. To enforce such constraint, it is sufficient to use as fit function

$$h_n(x) = f_n(x) - f_n(0) = \sum_{r=1}^{n} \tilde{\alpha}_r \left[ P_r(x) - P_r(0) \right]. \tag{A.9}$$

Finally, we reconstruct the original expansion in the naive basis by summing all equal monomials among every $P_r(x)$ included in the fit function:

$$h_n(x) = \sum_{r=1}^{n} \tilde{\alpha}_r \left[ P_r(x) - P_r(0) \right] = \sum_{r=1}^{n} \tilde{c}_r x^r, \tag{A.10}$$

where

$$\tilde{c}_r = \sum_{l=r}^{n} \tilde{\alpha}_l \left. \frac{d^r P_l(x)}{dx^r} \right|_{x=0}. \tag{A.11}$$

Once the two expansions are properly matched, the coefficients obtained from orthogonal polynomials agree with those obtained using a standard polynomial fit. The advantage of orthogonal polynomials resides in their improved numerical stability, which results in an improved precision in the computation of the $c_i$.

Finally, once the best fitting polynomial is obtained, we assign an error to our best fit function $h_n(x)$ through standard error propagation, via the so-called parameter covariance matrix,

$$C_{ij} \equiv \text{Cov}(\tilde{\alpha}_i, \tilde{\alpha}_j). \tag{A.12}$$

Let us define $v_i(x)$ as the gradient of the fit function with respect to the $i^{\text{th}}$ fit parameter,

$$v_i(x) = \frac{\partial h_n(x \,|\, \vec{\tilde{\alpha}})}{\partial \tilde{\alpha}_i}. \tag{A.13}$$

The error on the best fitting polynomial is

$$\text{Err}(h_n)(x) = v^{\text{T}}(x) C v(x) = C_{ij} v_i(x) v_j(x). \tag{A.14}$$

The best fit of $\gamma_{\mathcal{O}}(y)$ via orthogonal polynomial regression was done by using the `curve_fit` routine from the standard `Python` library `scipy`.

# B Resummation of perturbative series

## B.1 Toy model example

In this appendix, we discuss the perturbative expansion of a toy model in dimension zero:

$$\mathcal{I}(g) \equiv \int_{-\infty}^{\infty} \frac{\mathrm{d}x}{\sqrt{2\pi}} \, e^{-\frac{x^2}{2} - g x^4}. \tag{B.1}$$

Its perturbative expansion is

$$\mathcal{I}(g) = \sum_{n=0}^{\infty} a_n (-g)^n, \qquad a_n = \frac{(4n)!}{2^{2n}(2n)!n!} \underset{n\to\infty}{\sim} \frac{2^{4n}}{\sqrt{2\pi n}} \times n!. \tag{B.2}$$

The analytic continuation of the integral (B.1) from $\text{Re}(g) > 0$ to the full complex plane is given by a second-kind modified Bessel $K$-function:

$$\mathcal{I}(g) = \frac{1}{4\sqrt{\pi g}} e^{\frac{1}{32g}} K_{\frac{1}{4}}\left(\frac{1}{32g}\right). \tag{B.3}$$

Using the asymptotic behavior of $K_{\frac{1}{4}}(z)$ for $z \to \infty$, one sees that the exponential prefactor is canceled, and the series (B.2) recovered. Note that $\mathcal{I}(g)$ has a cut on the whole negative real axis, see Fig. 22.

In field theory, the divergent series is analytically continued without the knowledge of its exact expression. Let us explain the strategy on the example of integral (B.1). The basic idea [51] to obtain a convergent series out of Eq. (B.2), is to divide each term by $n!$, defining the *Borel transform* $\mathcal{I}_{\text{B}}(t)$ of the series. In a second step, one reconstructs the original series via an integral transform:

$$\mathcal{I}_{\text{B}}(t) \equiv \sum_{n=0}^{\infty} \frac{a_n}{n!}(-t)^n, \quad \mathcal{I}(g) = \int_0^{\infty} \mathrm{d}t \, e^{-t} \, \mathcal{I}_{\text{B}}(tg). \tag{B.4}$$

In our example we know the analytic expression in terms of the first-kind complete elliptic integral function

$$\mathcal{I}_{\text{B}}(t) = \frac{2 K_{\text{elliptic}}\left(\frac{1}{2} - \frac{1}{2\sqrt{16t+1}}\right)}{\pi \sqrt[4]{16t+1}}. \tag{B.5}$$

The Borel transform $\mathcal{I}_{\text{B}}(t)$ has a finite radius of convergence, denoted by $-t_{\text{bc}}$ (equal to $1/16$ in our example). As a consequence, the start of the branch cut is moved from $g = 0$ to $t = t_{\text{bc}} < 0$, see figure 22. Since the radius of convergence of $\mathcal{I}_{\text{B}}(t)$ is still finite, the integral transform (B.4

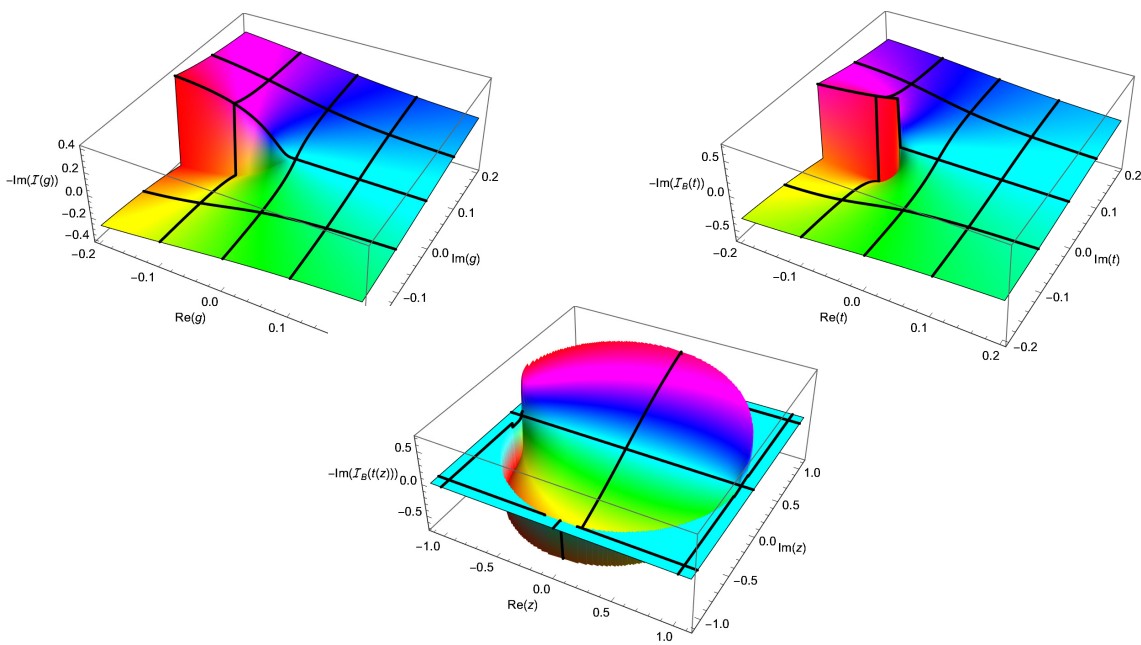

Figure 22: The branch cut in $\mathcal{I}(g)$ (top left) and $\mathcal{I}_B(t)$ (top right). While the former starts at $g = 0$, the latter is moved to $g = -1/16$. The lower plot shows $\mathcal{I}_B(t(z))$, which now has a branch-cut singularity at $|z| = 1$. (We set $\mathcal{I}_B(t(z))$ to 0 outside the disc $|z| \geq 1$.)

) does not work as written. One first has to continue $\mathcal{I}_B(t)$ to the domain $0 \leq t < \infty$. This can be achieved by replacing the known truncated series via a converging Padé approximant, leading to a *Padé-Borel resummation*.

A more powerful strategy is to use a conformal mapping. The most common ansatz is to assume that at $t = t_{bc} < 0$ a cut-singularity starts, which extends on the negative real axis to $t = -\infty$. One first maps the complex plane with the expected branch cut of $\mathcal{I}_B(t)$ onto the inside of the unit-circle:

$$z = \frac{\sqrt{1 - t/t_{bc}} - 1}{\sqrt{1 - t/t_{bc}} + 1} \qquad \Longleftrightarrow \qquad t = \frac{-4 t_{bc} z}{(z - 1)^2}. \tag{B.6}$$

Next one constructs a series in $z$ by expanding both sides in this variable:

$$f(z) \equiv \sum_{n=0}^{\infty} c_n z^n = \sum_{n=0}^{\infty} \frac{a_n (-t(z))^n}{n!} = \mathcal{I}_B(t(z)). \tag{B.7}$$

This series is expected to converge for $|z| < 1$, a fact we can check for our example (but which is difficult to prove in general):

$$f(z) = 1 - \frac{3z}{4} + \frac{9z^2}{64} - \frac{51z^3}{256} + \frac{1353z^4}{16384} - \frac{7347z^5}{65536} + \frac{61617z^6}{1048576} + \mathcal{O}(z^7). \tag{B.8}$$

Given $n$ terms in the original series, we know $f(z)$ up to the same order. Using this approximation for $f(z)$, we finally obtain:

$$\mathcal{I}(g) = \int_0^\infty dt\, e^{-t} \mathcal{I}_B(tg) = \frac{1}{g} \int_0^\infty dt\, e^{-t/g} \mathcal{I}_B(t) = \frac{1}{g} \int_0^1 dz\, t'(z) e^{-t(z)/g} f(z). \tag{B.9}$$

The result of this resummation is shown on Fig. 23. First, in black is the analytic result (B.3). Next are the first three orders in several expansions, using the same color code for order

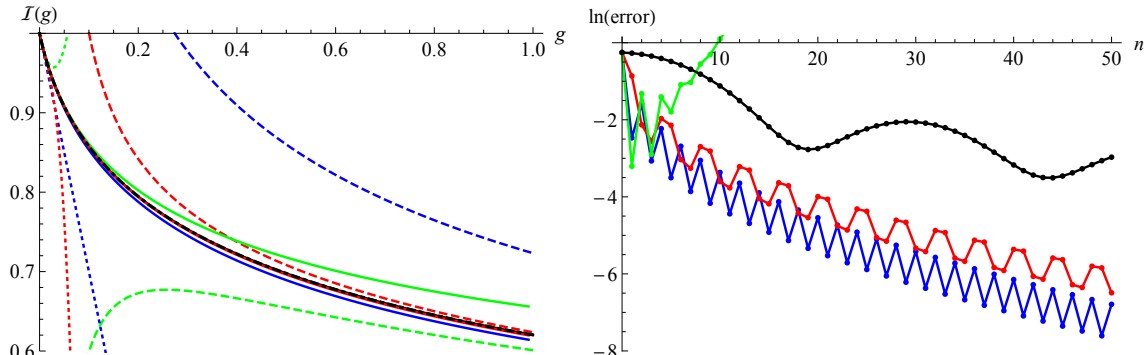

Figure 23: Left: function $\mathcal{I}(g)$ (black, thick, dot-dashed) and its diverse approximations. Dotted for the series expansion at order 1 (blue), 2 (green), and 3 (red). Solid for the resummed series at the same order. Dashed for the large-$g$ expansion (same color code). Right: deviation of the resummed series (B.9) from the exact result (B.5) for $g = 10$ as a function of $n$, assuming one knows $t_{\text{bc}}$ only approximately. In blue for $t_{\text{bc}} = -1/16$ (the exact result), in red $t_{\text{bc}} = -1/32$ (a conservative guess), in black $t_{\text{bc}} = -1/1000$ (much too small). Resummation with $t_{\text{bc}} = -1/15$ (green) does not work. We see that conform to expectations, taking a too small value for $-t_{\text{bc}}$, the series converges more slowly, while taking a too large value of $-t_{\text{bc}}$ the series does not converge.

1 (blue), 2 (green), and 3 (red): first the direct expansion in $g$ (dotted), then in solid the resummed expansion (B.9). Dashed, we show a large-$g$ expansion obtained by changing variables $gx^4 \to y$ in the integral (B.1), and then expanding the integrand in powers of $1/\sqrt{g}$:

$$
\begin{aligned}
\mathcal{I}(g) &= \frac{1}{2\sqrt{2\pi}\sqrt[4]{g}} \int_0^\infty dy \, \frac{e^{-\frac{\sqrt{y}}{2\sqrt{g}} - y}}{y^{\frac{3}{4}}} \\
&= \frac{1}{2\sqrt{2\pi}\sqrt[4]{g}} \left[ \Gamma\left(\tfrac{1}{4}\right) - \frac{2}{3} \frac{\Gamma\left(\tfrac{7}{4}\right)}{\sqrt{g}} + \frac{\Gamma\left(\tfrac{5}{4}\right)}{8g} + \mathcal{O}\left(g^{-\frac{5}{4}}\right) \right].
\end{aligned}
\tag{B.10}
$$

## B.2  Details on the resummation method

This appendix aims at providing a "reader's guide" to the analysis in Ref. [40], which determines the resummed series for the $d = 3$ critical exponents $\eta$, $\nu^{-1}$ and $\omega$ (related, respectively, to $\gamma_\sigma$, $\gamma_\epsilon$ and $\gamma_{\epsilon'}$). The same methods are used in our work, by a simple generalization to varying dimension $4 > d \geq 3$. This guide, together with the introduction in the main text and the example in App. B.1, should provide enough information to follow the discussion in Ref. [40]. In particular, we are interested in its Sec. V. Let us denote the equations in Ref. [40] by double parentheses, e.g., Eq. ((25)), to avoid confusion with our numbering.

The resummation procedure with Borel transform and conformal mapping goes along the lines described in our Sec. 3.2 and App. B.1. The perturbative series of a critical exponent $f(\varepsilon)$ in $(-2\varepsilon) = D - 4$ (cf. our $y = 4 - d$) is defined in Eq. ((25)) of [40]:

$$
f(\varepsilon) = \sum_{k=0}^{\infty} f_k \, (-2\varepsilon)^k, \qquad f_k \sim C_f \, k! \, a^k \, k^{b_f}, \quad \text{as} \quad k \to \infty.
\tag{B.11}
$$

With respect to our notation (cf. our Eq. (6)), the negative sign of $a$ is included in the power of epsilon, and the exponent of the power-law behavior earlier denoted by $b$ is now $b_f$.

The values for the parameters $(a, b_f)$ are given in Eq. ((26)) for the $\lambda\phi^4$ theory with O($n$) symmetry, $n = 1$ being the case of interest, and they are determined by the known asymptotic

behavior of the beta function. With respect to the definition given here in Eq. (11), in [40] the Borel transform is replaced by the more general Borel–Leroy transform, defined as follows (cf. Eq. ((27)) in [40]):

$$\mathcal{B}_f^b(x) = \sum_{k=0}^{\infty} \frac{f_k}{\Gamma(k+b+1)} (-x)^k \,, \tag{B.12}$$

where $b$ is a free parameter. The function $\mathcal{B}_f^b(x)$ behaves as $\mathcal{B}_f^b(x) \sim (1+ax)^{b-b_f-1}$ around $x = -1/a$.

The function $\mathcal{B}_f^b(x)$ is then modified in three ways in order to define the inverse transform and improve its convergence. The first step is the conformal mapping ((29)) already described in App. B.1, involving the known parameter $a$. The second step is the addition of the power-law prefactor in ((30)) with a second free parameter $\lambda$. The third step is the "homographic transformation" $\varepsilon = h_q(\varepsilon')$ defined in ((32)) which introduces a third free parameter $q$.

The resummed epsilon-expansion series $\tilde{f}(x)$ is finally obtained from the inverse Borel transform of the modified function $\mathcal{B}_{f\circ h_q}^{b,\lambda,\ell}$ reported in ((33)) of [40],

$$\tilde{f}(\varepsilon) = \int_0^{\infty} t^b e^{-t} \mathcal{B}_{f\circ h_q}^{b,\lambda,\ell}\left(\frac{2\varepsilon t}{1-q\varepsilon}\right) dt \,. \tag{B.13}$$

It depends on three free parameters: $b$, $\lambda$ and $q$ ($\ell$ being the perturbative order considered, $\ell = 6$ here). Let us briefly mention how these are determined.

The behavior of $\tilde{f}(\varepsilon) \equiv \tilde{f}_\ell^{b,\lambda,q}(\varepsilon)$ is studied in the cubic range

$$(b, \lambda, q) \in [0, 40] \times [0, 4.5] \times [0, 0.8] \,. \tag{B.14}$$

The optimal values of the parameters are chosen according to the principle of "minimal sensitivity" (w.r.t. varying the parameters) and "fastest apparent convergence" (w.r.t. increasing the perturbative order by one, $\ell - 1 \to \ell$). These dependences are taken into account by a proper definition of the error function $E_\ell^f(b, \lambda, q)$ that is given in Eq. ((36)).

The global minimum of the error $\bar{E}_\ell^f = E_\ell^f(\bar{b}, \bar{\lambda}, \bar{q})$ in the cubic range (B.14) identifies the optimal values $b = \bar{b}$, $\lambda = \bar{\lambda}$ and $q = \bar{q}$. The final estimate for the critical exponents is obtained from the inverse Borel transform (B.13) with these parameters. The optimization procedure is done independently for each dimension $d = 4 - 2\varepsilon$. The results for $\bar{b}$, $\bar{\lambda}$ and $\bar{q}$ are reported in Tab. 7 for the resummations of $\eta$, $\nu^{-1}$ and $\omega$ at the $d$ values considered. Note the mild dependence of the parameters on $d$.

We remark that this brief outline brushes over many fine details discussed in Ref. [40], but which are crucial for achieving high-quality results, as well as the comparison with other methods developed in the extensive literature. More technical information can be found in Ref. [40] and its supplementary material, available in arXiv:1705.06483.

The Self Consistent resummation of perturbative data used to obtain results shown in Figs. 10, 11, 12, 14 and 17, instead, does not involve the optimization of free parameters introduced before. As explained in Sec. III of [41], the asymptotic behavior (B.11) is fitted from the perturbative series, thus finding the position of the singularity $x = -1/a$ of the Borel transform. Such fit is done for several values of the free parameter $\alpha$, defined in Eq. ((44)) of [41] and analogous to $b_f$ in (B.11), varied in the range $-6 \leq \alpha < 6$ in steps of 0.2.

For each $\alpha$, the value of $a$ obtained from the fit is used in the conformal mapping (B.6) ($t_{bc} \equiv a$) and the resulting function is Borel inverted, giving the resummed series. The best estimate of the resummed quantity with this procedure is obtained through the mean over all the values of $\alpha$ employed, while the error bars represent the maximal and minimal values obtained varying $\alpha$. Since only one parameter is varied, this error estimate is less reliable than that determined by the methods of Ref. [40] described earlier.

Table 7: Optimal variational parameters used here in the resummation procedure for the critical exponents $\eta$, $\nu^{-1}$ and $\omega$, as a function of $4 > d \geq 3$.

|  | $d$ | $\bar{b}$ | $\bar{\lambda}$ | $\bar{q}$ |
|---|---|---|---|---|
| $\eta$ | 3.875 | 11 | 2.56 | 0.20 |
|  | 3.75 | 11 | 2.56 | 0.20 |
|  | 3.5 | 11 | 2.56 | 0.20 |
|  | 3.25 | 11 | 2.56 | 0.20 |
|  | 3 | 11 | 2.56 | 0.20 |
| $\nu^{-1}$ | 3.875 | 15 | 1.32 | 0.16 |
|  | 3.75 | 15 | 1.32 | 0.16 |
|  | 3.5 | 15 | 1.32 | 0.16 |
|  | 3.25 | 14 | 1.30 | 0.16 |
|  | 3 | 13.5 | 1.30 | 0.16 |
| $\omega$ | 3.875 | 19 | 0.52 | 0.46 |
|  | 3.75 | 21.5 | 1.02 | 0.40 |
|  | 3.5 | 21.5 | 1.02 | 0.40 |
|  | 3.25 | 22 | 1.02 | 0.40 |
|  | 3 | 22 | 1.02 | 0.40 |

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
