# Peer review of "Benchmarking the Ising Universality Class in $3 \le d < 4$ dimensions"

_SciPost Physics, doi:SciPost Phys. 14, 135 (2023)_

## Round 1 · Referee Report · Anonymous (Referee 1) · 2022-12-21

Strengths
1- Well-written manuscript 2- Improved estimates for many operator dimensions and structure constants for $3<d<4$ 3- Clear appendix with a worked example of Borel transform
Weaknesses
1- Bootstrap study of a well-known system with no significant novelty 2- Discussion on Borel transform and resummation in the main text lacks details 3- No discussion of systematic errors
Report
The paper concerns the $d=4-\epsilon$ expansion for the Ising CFT, and the aim of the paper is to compare predictions from this expansion with results from the numerical conformal bootstrap at finite $\epsilon$.
The method is a bootstrap scheme consisting of bounds from a single correlator together with central charge minimization, followed by an extraction of the extremal spectrum. This gives access to the leading critical exponents and a set of CFT-data (dimensions and structure constants with sigma) for low-lying operators. Errors are estimated by comparing the position of the kink in the bound and the position of the minimum value of the central charge.
The main conclusions are that the unresummed $\epsilon$ expansions give good agreement for $3.8<d<4$, and that resummed expansions give good agreement in the whole studied range $3\leq d<4$. The paper also gives a collection of estimates for CFT-data that is a nice contribution to the literature.
The manuscript is well-written and conveys its main results in a clear way. However, I have doubts about the novelty of the work, for the following reason:
The bootstrap method using a single correlator is a direct application of the method used by some of the authors in 2018 (ref. 19). But more refined techniques have been available for a while and applied to the Ising CFT. Specifically, in the range $3<d<4$, bootstrap islands have been known since 2016 (see 1602.02810; the authors may choose to include this paper in their reference list). The only novelty with respect to the bootstrap method is a denser sampling of points near the kink, and an additional point $d=3.875$. Also the estimates from resummation appear to be a direct application of previously developed techniques, however there is a lack of details that makes it hard to judge the potential difficulties behind the produced estimates.
Based on this, I make the judgement that the current manuscript does not meet the acceptance criteria of the journal, section "Expectations". Since the results of the study may still be a useful addition to the literature, I would reconsider my recommendation after major revisions. One way to meet the criteria would be to make a more elaborate discussion on the resummation methods, which are not given enough room in the current draft. By expanding on that aspect, the paper could become a useful resource for future projects comparing perturbative series and non-perturbative results.
1) Could you give some more details on how the resummation was computed? How were the errors estimated? What value does the parameter $a$ take? Which variational parameters were used? Note the “General acceptance criterion” number 5 of this journal.
2) You mention another resummation technique (hypergeometric resummation of ref. 55). Could you compare against this technique and give recommendations based on the comparison? Also, it appears like the self-consistent resummation for $3<d<4$ was only performed for $\Delta_\epsilon$ and not the other quantities.
3) Footnote 8 comments on the inclusion of seventh-order results, saying that they give larger errors than sixth-order results. Could you provide more details?
I also have the following remarks/questions:
4) Comparing with the "Navigator" of ref. 20 (red triangles in the draft) there appear to be systematic errors of the same size, or larger, than the errors given. This is the case at least in figures 4 and 17. Could you discuss the origin of these systematic errors and, possibly, give a method to estimate their size?
5) What does 190 components correspond to in terms of the value $\Lambda$ that is commonly given in the bootstrap literature?
6) In table 1, comparing the size of the errors in $d=3.875$ and $d=3.75$, precision for spin 4 is much higher in $d=3.875$, and for spin 0 and spin 2 much higher in $d=3.75$. Is there any explanation for this?
7) In appendix B, you mention that convergence for $|z|<1$ is difficult to prove. Does it not follow from the assumptions made on the analytic structure in the $t$ plane?
Requested changes
See points 1) to 7) in the report

---

## Round 1 · Referee Report · Anonymous (Referee 2) · 2023-1-19

Report
The paper by Bonanno, Cappelli, Kompaniets, Okuda, Wiese [BCKOW] studies the Wilson Fisher fixed point in d=4-epsilon dimensions for various values of epsilon using the conformal bootstrap and compares the results to other methods.
The paper adds 4 grid points in d in the range 3<d<4 (d=3.875, 3.75, 3.5, 3.25) to the 10 grid points for 2<d<3 already computed in a previous reference [19]. For each value of d, various conformal dimensions and OPE coefficients are estimated. For each data point a non-rigorous estimate of the error is also provided. The computation is done with a single correlator bootstrap setup (with c-minimization) which the authors themselves admit “has been surpassed by more recent implementations”.
The main result is a fit in d of the conformal bootstrap observables. The paper also provides many checks of the consistency of the fits with other techniques as resummation of the perturbative series, Monte Carlo simulation and other bootstrap approaches.
The paper is well written, clear and the results are interesting. However unfortunately, reference [20] appeared a few months before with the same type (but better quality) of bootstrap results. In particular [20] studies the following values of d: d=2.6, 2.7, 2.8, 2.9, 3.0, 3.1, 3.2, 3.3, 3.4, 3.5, 3.6, 3.7, 3.8, 3.9, 3.95. It thus gives 10 new data points in the range 3<d<4 (instead of only 4). Moreover the results of [20] are obtained with a much more sophisticated technique (one of the “more recent implementations” above), which are expected to be far more precise: [20] is in fact using a multiple-correlator setup combined with the recently introduced navigator method. While [20] is believed to be far more accurate than any single-correlator bootstrap, admittedly the authors of [20] do not estimate the errors in their data points so a honest comparison is not easy to perform (this is also hard since there is no way to know if the estimated error in [BCKOW] is correct). However some plots of [BCKOW] clearly show that the bootstrap results of [BCKOW] are far less precise than [20]. For example in figure 20 it is shown that [20] finds two operators which lie (with good agreement) on the epsilon expansion results, while [BCKOW] only resolves a single operator which interpolates between the two curves. One should also stress that when there are discrepancies between the results of [BCKOW] and [20], there is a good reason to prefer the ones of [20]. Indeed the latter are by construction inside an allowed island provided by the mixed correlator setup, while the estimates of [BCKOW] together with their full uncertainty can possibly live in the disallowed region and so they could be mathematically ruled out.
Because of the existence of [20], the bootstrap results of [BCKOW] are outdated. However they may be still instructive to show the power/precision of the single correlator bootstrap, which is much simpler to implement. [BCKOW] also give estimates for some OPE coefficients which are not presented in other works. The paper also provides fits of the bootstrap data which may be of interest. The thorough comparison with the resummation techniques is also a valuable addition to the literature. I would thus recommend [BCKOW] for publication under major revision.
My main problem with the paper is that it is using outdated bootstrap results treating them on the same footing as the new state of the art results of [20]. A reader who is not expert in the bootstrap may likely think that the bootstrap results of [BCKOW] are comparable or even more accurate than [20]. This is misleading. In particular I do not agree with the choice of phrasing the paper as a comparison of many techniques the bootstrap results of [BCKOW]. The paper would be more useful if the comparison was made with the new data of [20]. This means that the authors could provide a polynomial fit of the data of [20] and use that to define the origin of their plots. They could then plot their 1-correlator results and show that these are close enough to the origin given by the fit of [20]. The broad logic of the paper (which should be stated from the abstract to the conclusion) should be: - provide a fit of the bootstrap results of [20] and compare it to other methods like resummation, Monte Carlo, etc. - provide a fit of the much simpler 1-correlator bootstrap and show that this is not too far from [20], thus showing that this technique could be used in other cases where the results of [20] are not yet available
In order to achieve this logic I propose the following changes:
1) Write a fit of the data of [20] for all observables for which this is available. For example formula (10) should be also compared with a new formula for a polynomial interpolating the data of [20]. Similarly for all other fits.
2) Change the origin of all plots to the one provided by the fit of [20] (when available).
3) Throughout the paper it should be made clear that the 1-correlator results are not the best ones available in the literature, but that they are simpler to achieve. For example in lines 91-95 only d=3 implementations are mentioned, while it should be stated that reference [1602.02810] already used a mixed correlator setup in fractional dimensions and that [20] uses mixed correlator with the navigator, thus surpassing the bootstrap results of [BCKOW]. The fact that the bootstrap results of [BCKOW] are surpassed should also be clear from the introduction and the conclusions where the works of [1602.02810] and [20] are not even mentioned in the current version.
4) In figure 4 (and similar) we can see that the data of [20] lies consistently below the shaded area of the fit of [BCKOW]. This suggests that the bootstrap data points of [BCKOW] produced by c-minimization have a systematic error, which is not taken into account in the paper. If it is possible it would be useful to estimate such error. An option is to simply increase the error in order to contain the points predicted by [20] (this is also a safe measure to be sure that future rigorous bootstrap bound will not completely rule out the 1-correlator fits of [BCKOW]). If these changes are not implemented, at least it should be stated that the method likely suffers from a systematic error which is hard to estimate.
Other comments/questions follow:
5) Since one of the most valuable parts of the paper is the comparison with resummation techniques, it would be useful to expand on the latter. Can you review the details of the various resummation methods used in the paper and how the errors are estimated?
6) Did you try to use other non-polynomial fits of the bootstrap data? For example, inspired by the Padé approximations of the perturbative series, one could try rational functions. It would be instructive to know if other fits have advantages.
Requested changes
See points 1)-6) above.

---

## Round 2 · Referee Report · Anonymous (Referee 2) · 2023-1-30

Report

The authors used a different, more minimal, strategy to address my questions. I am fine with the new strategy and I believe that the resulting paper is worth to be published with minor revisions.

First let me comment on the answer to my point 1).
I agree that "a fit of data without errors is only
qualitative", but one should also take into account that the estimate of the error given in the paper is completely non rigorous, since it is based on unverified assumptions. E.g. it would not be mathematically inconsistent if some predictions were off by a factor of two, this would only be unexpected by the bootstrap lore that theories live close enough to kinks in the exclusion plots (notice however that this lore sometimes fails). So the method itself and the estimate of the error are more like an art than a science.
For this reason I still believe that the "qualitative" fit of the data of [21] is more precise and mathematically more justified than the one's of the authors. Indeed the data of [21] actually has a mathematically rigorous (and probably very small) error, even if unfortunately this was not estimated in the paper.
Said so, I agree with the point of the authors that, since the main scope of the paper is to compare with resummation, and the latter has much larger errors, the improved precision of [21] is not needed. Also the fact that the method used in the paper compares well with [21] is interesting since it says that the bootstrap lore is correct within some uncertainty.

I only ask for a single minor revision.
I do not understand the choice of plotting Figs. 7, 9 and 13 including only a single point at d=3.5 from reference [21], while data is available for d=3.0, 3.1, 3.2, 3.3, 3.4, 3.5, 3.6, 3.7, 3.8, 3.9, 3.95. Why one point and why d=3.5?
I would add all possible points since they are available. If the problem is only at the level of clarity of the plot (because the red triangles are big and there are many of them), the authors could plot the curve that interpolates the data of [21] (even without writing the functional form of such curve), which would in my opinion would be a better reference "to assess the negligible difference between the two sets of bootstrap data in the comparison to the epsilon-expansion".

---

## Round 2 · Referee Report · Anonymous (Referee 1) · 2023-1-31

Report

The authors have successfully addressed my comments 4, 5 and 6. I also accept the motivation in the reply to comment 2 of not including the hypergeometric resummation.

Regarding comment 3, I understand the explanation for why the seventh-order results are not included in the draft, but I think it would be useful to provide more information in the draft. For instance, footnote 10 could be expanded with estimates in $d=3$ that compare the six-loop and seven-loop resummations for the three main critical exponents considered, giving the central value and uncertainty for each. This would give the reader a chance to examine the choice of limiting to six-loop results in the rest of the paper.

My comment 1 remains not addressed in the new version of the draft. The reference to [38] is useful but does not make the draft more self-contained. The specific value of the parameter $a$ and the choice of variational parameters cannot be found in the draft.

The authors note that also the section on the numerical bootstrap is rather short. This is true, but that section is suitably balanced toward the aspects that are specific to the paper at hand. The general theory is very briefly introduced, but the algorithm is clearly explained with reference to available software. The specific computations for the present paper (estimating errors based on the position of the kink and the central charge minimum, and the fitting of interpolating polynomials) are presented with sufficient details, and figures 1 and 2 are useful for the reader who wants to reproduce intermediate results.

On the other hand, the description of resummation methods is rather heavy on the toy model example, and provides almost no details on the adaptations of the general methods to the paper at hand. No intermediate results are given that make it possible to follow the computation. Neither does the draft have any associated computational files or software, or clear references to where such software can be found.

As it stands, my judgement is that the draft does not meet the general acceptance criteria, especially point 5. Without more details on the resummation methods, it is also difficult to see how the draft can meet any of the criteria in the section Expectations. Unless the authors submit a revised manuscript, I cannot recommend publication.

---

## Round 2 · Author Response

Warnings issued while processing user-supplied markup:

  • Inconsistency: Markdown and reStructuredText syntaxes are mixed. Markdown will be used.
    Add "#coerce:reST" or "#coerce:plain" as the first line of your text to force reStructuredText or no markup.
    You may also contact the helpdesk if the formatting is incorrect and you are unable to edit your text.

Dear editors,

We thank the referees for the careful reading of our manuscript, and for their comments. We are resubmitting our paper with a fair number of modifications, following the referees' suggestions. However, we did not consider a major rewriting, and in the following we would like to motivate this choice. More specific answers to queries follow afterwards.

=======

First referee.

A major revision was requested for better explaining the resummation methods. We think that extensive additions are not necessary, because:

i) there is a huge literature on this historically established subject. In our paper we provide a rather simple introduction in Sect. 3.2 and discuss a toy model in appendix A; in the revised version, we address the reader to the paper by one of the authors, Ref. [38], where a specific resummation, very close to the one used in our paper, is nicely presented. After these readings, the main Refs. [40-41] for our work should be directly accessible.

ii) the aim of our paper is the comparison of two rather different research topics: numerical bootstrap and perturbative expansion. Attempts to reviewing either method would end up in a very lengthy paper: actually, while the bootstrap approach is much less explained than resummation methods, the referees only asked on details of the latter.

  • Corrections implemented: We added remarks in several parts of the paper, for pointing to review material, and for better motivating the used methods. See later for answers to specific questions on this issue (and the separate list of corrections).

==========

Second referee.

The request a for major revision was motivated by the relation of our work with Ref. [21] (Ref. [20] in the previous version of the manuscript), presenting results obtained by the 3-correlator bootstrap with the navigator method. The Referee argued that these results are more precise than ours, that our approach is outdated, and that we should rewrite completely our analysis using the data from Ref. [21]. However, we believe that a major rewriting of our paper is not necessary:

i) The data of [21] are certainly very precise, but unfortunately they come without error estimates, so they cannot be directly used for fits, which are the baseline for our comparison with perturbative methods. Furthermore, results for the structure constants were not provided in [21], while our data are even more precise than those for conformal dimensions.

ii) We would like to point out that while the data of Ref. [21] may be more precise than ours, it is always good to have an independent verification, and different questions asked. We view Ref. [21] and our paper as complementary in this respect. Actually, in our paper we show a rather satisfactory agreement in the low-lying part of the conformal spectrum: the data agree, up to very tiny differences. The disagreement for higher-dimensional states was discussed in Sect. 4.3, where we made clear that the navigator results are better.

Corrections implemented:

  • we gave more credit to Ref. [21] and the navigator method in several parts of the revised paper, including Introduction, Sect. 2.1, and Conclusions. We said that our method is inferior, yet useful for the problem at hand, since it is computationally much cheaper.

  • we updated the comparison of data for the two approaches in Fig. 4: we propose an estimate for the navigator errors, observed the small offset in our data, remarked that the navigator approach correctly identifies the unitarity island. We then enlarged our error estimates to account for this offset (see gray areas in the plots of Fig. 4).

  • we added navigator data to Figs. 7, 9 and 13, where the comparison with resummed perturbative results is made. This allows us to estimate the potential error of our fits.

  • we added the reference to earlier work [1602.02810]; we are sorry that we forgot to mention it, but actually its data have errors too large to be used in our fits.

=============

Answers to referees' questions

  • First referee

1) Could you give some more details on how the resummation was computed? How were the errors estimated? What value does the parameter a take?

Answer was given above. The parameter $a$ is defined by the asymptotics (3.2); the instanton calculus provides a semiclassical estimate.

2) You mention another resummation technique (Hypergeometric resummation of ref. 55). Could you compare against this technique and give recommendations based on the comparison? Also, it appears like the self-consistent resummation for $3<d<4$ was only performed for $\Delta_\epsilon$ and not the other quantities.

The use of Hypergeometric functions is yet at a primitive stage; in our paper, we did not discussed it, because a whole paper needs to be written before its precision could properly be addressed; we added a short sentence on page 13.

As explained in Ref. [41], the self-consistent resummation gave small errors only for the quantity $1/\nu^3$, which has a good limit for $d\to 2$ into an exactly-known expression. A similar strategy was not found for the other exponents. We added a short sentence on page 15.

3) Footnote 8 comments on the inclusion of seventh-order results, saying that they give larger errors than sixth-order results. Could you provide more details?

The seventh-order perturbative terms have not been independently confirmed, and there are some concerns about their validity in the community. Attempts by two of the authors to use them in the resummations were not satisfactory. Here too we prefer to be rather brief, to not convey any criticism or indulge into speculation.

4) Comparing with the "Navigator" of ref. 20 (red triangles in the draft) there appear to be systematic errors of the same size, or larger, than the errors given.

This point was answered above and is now discussed in the revised version.

5) What does 190 components correspond to in terms of the value $\Lambda$ that is commonly given in the bootstrap literature?

It corresponds to $\Lambda = 18$. We added a footnote in Sec. 2.1 to clarify this point.

6) In table 1, comparing the size of the errors in d=3.875 and d=3.75, precision for spin 4 is much higher in d=3.875, and for spin 0 and spin 2 much higher in d=3.75. Is there any explanation for this?

In the 1-correlator bootstrap, the states higher up in the conformal spectrum change rapidly within the range of parameters identified as the Ising point (see Sect. 2.1). The low-lying operators, more precisely the leading twists, are more stable. This is a known fact. In the case of $\Delta_{\epsilon'}$ and $\Delta_{\epsilon''}$ this instability is parametrically sharper when passing from $d=3.75$ to $d=3.875$, thus explaining the increase in errors. Instead, the spin-four field $C$ is a leading twist and is not affected. We added a footnote on page 20.

7) In appendix B, you mention that convergence for $|z|<1$ is difficult to prove. Does it not follow from the assumptions made on the analytic structure in the $t$ plane?

The problem is that the analytic structure in the $t$ plane in not known for general field theories. In the toy model discussed in App.B, analytic properties are explicit.

Second referee

1) Write a fit of the data of [20] for all observables for which this is available...... 2) Change the origin of all plots to the one provided by the fit of [20] (when available).

Already replied: in general, a fit of data without errors is only qualitative; nevertheless, in the revised versions we added navigator data to our fits for comparison.

3) Throughout the paper it should be made clear that the 1-correlator results are not the best ones available in the literature, but that they are simpler to achieve.....

Already replied. The suggested modifications have been implemented.

4) In figure 4 (and similar) we can see that the data of [20] lies consistently below the shaded area of the fit of [BCKOW]. This suggests that the bootstrap data points of [BCKOW] produced by c-minimization have a systematic error, which is not taken into account in the paper. If it is possible it would be useful to estimate such error. An option is to simply increase the error in order to contain the points predicted by [20]........

Already replied. This issue of the offset is discussed in the revised version; it is now included in the error estimates.

5) Since one of the most valuable parts of the paper is the comparison with resummation techniques, it would be useful to expand on the latter. Can you review the details of the various resummation methods used in the paper and how the errors are estimated?

This point was also raised by the first referee and it is answered above.

6) Did you try to use other non-polynomial fits of the bootstrap data? For example, inspired by the Pade' approximations of the perturbative series, one could try rational functions. It would be instructive to know if other fits have advantages.

Pade' approximations are used for representing functions that have a finite radius of convergence. As explained in Sect. 3.1, after (3.6), the bootstrap non-perturbative data we are approximating by polynomials are expected to be analytic functions in dimension $d<4$, and to have a branch cut singularity for $d>4$. A Pade' approximant could not properly model this.

Best regards,

The Authors (Bonanno, Cappelli, Okuda, Kompaniets, Wiese)

---

## Round 2 · List of Changes

• Sec. 1, 3rd paragraph, added

"The conformal spectrum for varying dimension 4 > d >= 2.6 has also been obtained in Ref. [21] by using the more advanced bootstrap technique of the navigator method [37]. We use these very precise results in combination with ours to obtain a consistent description of the low-lying spectrum."

  • Sec. 1, 5th par., rewritten

"The analysis is done on the dimensions of the conformal fields sigma, epsilon, epsilon^prime, respectively corresponding to spin, energy and subleading energy, which determine the critical exponents eta, nu, omega. The precision of bootstrap data can be summarized by the d-independent value of the relative error Err(gamma)/gamma = O(10^(-3) ) for the anomalous dimensions gamma for the conformal fields sigma, epsilon. As the anomalous dimensions are very small for d \approx 4, the precision for the conformal dimensions delta_sigma, delta_epsilon is actually higher in this region. Regarding the subleading energy, the relative error Err(gamma_epsilon^\prime)=gamma_epsilon^\prime stays at three digits, as explained later. Furthermore, some of the structure constants are determined with even better O(10^4) accuracy."

  • Sec. 2.1, 1st par. added footnote

"This corresponds to the parameter Lambda = 18 counting the number of derivatives in the approximation of the functional basis."

  • Sec. 2.1, 2nd par., added references [16, 45, 46] -> [16, 19, 21, 47, 48] and rewritten

"Our 1-correlator numerical bootstrap approach has been surpassed by more recent implementations [16,19,21,47,48], but we find it convenient for determining the low-lying spectrum with modest computing resources."

  • Sec. 2.2, last 3 pars., rewritten

"Next, we compare these results with those recently obtained by solving the 3-correlator bootstrap with the navigator method [21]. In Fig. 4 our data, given in earlier figures (blue circles), are shown on a finer scale, together with the estimated error of the fit (cyan shaded area). The red triangles are the navigator values: they come with no errors and thus cannot be directly used for the fits. A first observation is the fairly good agreement between the two different bootstrap approaches at our level of precision.. We propose to estimate the error of navigator data as follows. We suppose that they are roughly of the same size as those found in other 3-correlator studies at d = 3 (rigorous bounds) [48, 50], which are plotted in Fig. 4 as black diamonds (gamma_sigma and gamma_epsilon), and a grey rightward triangle (epsilon^prime). Assuming these very small uncertainties for each value of d, there seems to be a negative offset with respect to our data, in particular for "0. This could be a systematic error due to our approximate identification of the Ising point within the unitarity region (Section 2.1), while the navigator method rigorously determines it within a unitarity island [37]. However, other explanations are possible.

In conclusion, taking into account these considerations, we enlarge the error estimate of our fits to the shaded gray bands in Figs. 4, which correspond to the following bound:"

  • Sec. 2.2, last par., added footnote

"Earlier results of Ref. [19] are not considered here due to their large errors."

  • Sec. 2.2, Fig. 4 modified by adding new error bounds (in gray)

  • Sec. 3.2, 4th par., added

"In our work, the resummed data are obtained by extending the setup of Refs. [40, 41] to noninteger dimension. A complete account of these methods is too long to be presented here: nonetheless, our introduction, App. B and the paper [38] provide enough background for accessing the original work."

  • Sec. 3.2, 5th par., added

"As here we could not give justice to their influence, we exclude this resummation method."

  • Sec. 3.2, 6th par., added

"Finally, Fig. 7 and later plots comparing the dimensions delta_epsilon, delta_epsilon^prime also report the result of navigator bootstrap for d = 3:5 (red triangle). This allows to assess the negligible difference between the two sets of bootstrap data in the comparison with the epsilon-expansion.

  • Fig. 7, 9, 13 modified by adding navigator point at d=3.5, and added to the caption

"Results from earlier works [3] have been omitted due to their large error bars."

  • Sec. 3.2, 9th par., added

"A similar constraint does not seem to be possible for the other critical exponents, as discussed in Ref. [41]."

  • Sec. 3.2, 13th par., added sentence

"The red triangle at d = 3.5 again indicates the offset with respect to the navigator bootstrap data."

and footnote

"The growth of the error when passing from d = 3.75 to d = 3.875 is due to the instability of the higher part of the spectrum when approaching d = 4. This issue will be further discussed in Sect. 4.3."

  • Sec. 4.2, 3rd par., modified

"Data from the navigator method are unfortunately only available for f_{sigma sigma epsilon} [21]."

  • Sec. 4.3, 1st par., added

"which are definitely more accurate for the higher spectrum than our results."

  • Sec. 5, 2nd par., added

"For these low-lying states of the conformal spectrum, our results are in very good agreement with those of more advanced 3-correlator bootstrap techniques [21, 37, 47, 48], with a small offset included in the error estimate."

  • Sec. 5, 5th par., rewritten

"We were able to compute bootstrap data for the conformal dimensions of higher-order fields in 4 > d >= 3, including the lowest-lying spinful fields T^prime (l = 2) and C (l = 4), with a precision comparable to that of spinless operators. The central charge and OPE coefficients of low-lying fields were obtained with even higher precision than that of the corresponding anomalous dimensions. The structure constants agree well with those of the 3-correlator bootstrap, where available (mostly in d = 3), and with perturbation theory for d -> 4."

---

## Round 3 · Referee Report · Anonymous (Referee 2) · 2023-2-15

Report

The authors answered all my questions and I believe that the paper is ready to be published.

---

## Round 3 · Referee Report · Anonymous (Referee 1) · 2023-3-6

Report

I thank the authors for the new version of the manuscript. The new appendix B.2. provides exactly the type of information that I was seeking in my previous requests. I also accept the decision about the treatment of the seven-loop results.

The paper may now be published if the following minor remark would be addressed: In the new appendix B.2., the relevant parameters for the resummations are given for $\eta$, $\nu^{-1}$ and $\omega$, but not for $f_{\sigma\sigma\epsilon}$. Could you report the values of $a$, $b_f$, $\bar b$, $\bar \lambda$ and $\bar q$ that were used to produce the estimates in figure 17?

---

## Round 3 · Author Response

Dear editors,

We thank the referees for their comments. We are resubmitting our
paper with an addition required by the first referee and some
modifications suggested by the second referee.

{\bf Answers to referees' requests}

{\bf First referee (report on 2023-1-31, revised version)}

{\it 1) (cf. question (3) of earlier report) Regarding comment 3, I
understand the explanation for why the seventh-order results are not
included in the draft, but I think it would be useful to provide
more information in the draft. For instance, footnote 10 could be
expanded with estimates in d=3 that compare the six-loop and
seven-loop resummations for the three main critical exponents
considered, giving the central value and uncertainty for each. This
would give the reader a chance to examine the choice of limiting to
six-loop results in the rest of the paper.}

As said in earlier reply, the seventh-order perturbative terms have been
obtained by Schnetz, but were not confirmed by other authors, and there are
some concerns about their validity in the community. Let us look at
the record: sixth-order calculations were independently checked by
Panzer, Kompaniets and Schnetz; fifth-order results were originally
incorrect and had to be modified by subsequent works.

On this basis, we cannot consider perturbative results of such a
complexity which have not been confirmed independently.
Using these data would make us prone to an erratum, while the gain,
assuming correctness, seems minimal.
In footnote 10 we were
rather vague in order to not offend Schnetz; we do not want to say that his
results are incorrect, simply that they need an independent check. We
modified the footnote 10 in a more explanatory way.

\bigskip

{\it 2) On the other hand, the description of resummation methods is
rather heavy on the toy model example, and provides almost no
details on the adaptations of the general methods to the paper at
hand. No intermediate results are given that make it possible to
follow the computation. Neither does the draft have any associated
computational files or software, or clear references to where such
software can be found.}

The resummation methods used in our work are the direct adaptation
to varying $d$ of the analyses in Ref. [40] and [41]. It is not possible
to do justice of the many aspects of these procedures in a short presentation
that would fit in our paper.
Furthermore, Sec. V of Ref. [40] offers a very clear and detailed
description: after the introduction in our paper, the reader can
easily follow all the steps there.

At any rate, in Appendix B.2 we added a survey of Sec. V of Ref. [40],
which can help the reader to find his way in this reference, including
main definitions and main steps, correspondence of notation, etc. We
also added the list of values for the variational parameters entering
the resummation process for any $d$ value considered.
Concerning more technical information, we address the reader to
the supplementary material of Ref. [40] that is available on arXiv:1705.06483.

\newpage

\bigskip

{\bf Second referee (Report on 2023-1-30, revised version)}

{\it I do not understand the choice of plotting Figs. 7, 9 and 13
including only a single point at d=3.5 from reference [21], while
data is available for d=3.0, 3.1, 3.2, 3.3, 3.4, 3.5, 3.6, 3.7, 3.8,
3.9, 3.95. Why one point and why d=3.5? I would add all possible
points since they are available. If the problem is only at the level
of clarity of the plot (because the red triangles are big and there
are many of them), the authors could plot the curve that
interpolates the data of [21] (even without writing the functional
form of such curve), which would in my opinion would be a better
reference "to assess the negligible difference between the two sets
of bootstrap data in the comparison to the epsilon-expansion".}

We put the $d=3.5$ point in Figs. 7, 9 and 13 because it corresponds
to the larger difference between navigator results and our fit, for all
the three critical exponents.
We did not put all the navigator points because they would
have messed up the plots.

Following the referee's suggestion, we put the line linearly interpolating
navigator data in these three figures. Now the differences between the
two sets of bootstrap data are clearer. We also added a remark after
Fig. 13, saying that the resummed perturbative data match
better the navigator results than our ones for $4>d \ge 3.5$.

Best regards,

The Authors (Bonanno, Cappelli, Okuda, Kompaniets, Wiese)

---

## Round 3 · List of Changes

• Sec. 3.2 - 4th paragraph - sentence modified

"A complete account 303 of these methods is too long to be presented here: nonetheless, our introduction, App. B and the 304 paper [38] provide enough background for accessing the original work."

->

"A complete account of these methods is too long to be presented here; nonetheless, we provide some introductory material that will allow the reader to assess the original works. In App.~\ref{appendix:baby_integral}, the resummation is worked out in a toy model, where one can compare it with the exact result. In App.~\ref{app:KP17_details}, instead, a ``reader's guide'' to Ref.~\cite{panzer} is presented, together with the values of the resummation parameters used here."

  • Sec. 3.2 - footnote 10 modified

"Resummations in this section use the 6th-order expansion that received several checks. Contrary to expectation, the apparent error at 7-loop order seems to be larger than that at 6-loop order, in all resummation schemes we tried [40, 41]."

->

"Resummations in this section use the $6$-loop results, that were verified in several independent works~\cite{panzer,Kompaniets:2019zes,Schnetz:2016fhy}. We do not use the $7$-loop results of Ref.~\cite{Schnetz:2016fhy}, since they were not yet checked independently. Past experience, e.g., with the $5$-loop results, teaches us that involved perturbative calculations require confirmation."

  • Figs. 7, 9, 13 modified + added a sentence in the caption

"We also plot a solid red line linearly interpolating results of Ref.~\cite{Henriksson:2022gpa} for $4>d\ge3$."

  • Sec. 3.2 - last paragraph - sentence modified

"A drift towards lower values for the green epsilon-expansion points is seen, as for $\gamma_\epsilon$. "

->

"A systematic difference between bootstrap and epsilon-expansion points is seen for $d \to 3$, similar to what was found for $\g_\e$ in Fig.~\ref{fig:e-eps}. Such a drift is smaller for the navigator results~\cite{Henriksson:2022gpa} (red line) than for our data, for $4 >d \ge 3.5$."

  • Added new Appendix B.2 (old Appendix B is now Appendix B.1)

---

## Round 4 · Author Response

``The paper may now be published if the following minor remark would be
addressed: In the new appendix B.2., the relevant parameters for the
resummations are not given for f_{\sigma\sigma\epsilon}. Could you
report the values of a, b_f, \bar b, \bar\lambda and \bar q
that were used to produce the estimates in figure 17?''
In Figs. 17 we used the Self-Consistent resummation method for f_{\sigma \sigma \epsilon} described
in Ref. [41], that does not involve the same optimization of parameters.
At the end of App. B.2, we have added a brief description of
this procedure, as a guide to reading Ref. [41], and compared with
that of Ref. [40] described earlier.
addressed: In the new appendix B.2., the relevant parameters for the
resummations are not given for f_{\sigma\sigma\epsilon}. Could you
report the values of a, b_f, \bar b, \bar\lambda and \bar q
that were used to produce the estimates in figure 17?''
In Figs. 17 we used the Self-Consistent resummation method for f_{\sigma \sigma \epsilon} described
in Ref. [41], that does not involve the same optimization of parameters.
At the end of App. B.2, we have added a brief description of
this procedure, as a guide to reading Ref. [41], and compared with
that of Ref. [40] described earlier.

---

## Round 4 · List of Changes

Added two paragraphs at the end of Appendix B.2.

---

## Editorial Decision

published